# Reasoning Planning for Language Models

**Bao Nguyen**[1]     **Hieu Trung Nguyen**[1]
**Ruifeng She**[2]     **Xiaojin Fu**[2]     **Viet Anh Nguyen**[1]

[1] The Chinese University of Hong Kong
[2] Huawei Noah's Ark Lab
nbnguyen@se.cuhk.edu.hk, thnguyen@se.cuhk.edu.hk
she.ruifeng@huawei.com, fuxiaojin32@hotmail.com, nguyen@se.cuhk.edu.hk

## Abstract

Selecting an appropriate reasoning method for a given query remains a key challenge in language model generation. Existing approaches typically generate multiple candidate responses and use an aggregation strategy to select the output answer, often assuming that more candidate answers yield higher accuracy. We revisit this assumption through a rigorous theoretical analysis, deriving accuracy bounds for standard aggregation methods under fixed generation distributions and candidate sizes. Building on these insights, we introduce EPIC, an **E**nsemble **P**lann**I**ng with **C**ontrastive learning framework to learn a shared representation space that captures both model reasoning abilities and query-method compatibility. EPIC incorporates our probability bounds as a regularizer in a utility-driven optimization that balances accuracy and computational cost. Experiments on diverse mathematical reasoning tasks show that EPIC consistently selects optimal reasoning methods, improving accuracy while reducing computational overhead. Our code can be found at `https://github.com/nguyenngocbaocmt02/EPIC`.

## 1 Introduction

Large Language Models (LLMs) have demonstrated remarkable abilities to understand and reason in human natural language. These advancements have transformed applications, including travel planning [Xie et al., 2024a], AI teaching platforms [Jin et al., 2024], and human population simulations [Park et al., 2023, Bui et al., 2025]. However, even with a pre-trained LLM, the computational expense of serving LLM-powered systems remains a significant bottleneck due to the massive scale of the models [Lin et al., 2024], the quadratic complexity of the attention mechanism [Dao et al., 2022], and the token-by-token nature of auto-regressive generation [Zhang et al., 2025]. This high computational cost significantly hinders the broader application of LLMs in practical scenarios, particularly in resource-constrained environments such as edge devices, real-time applications, and small-scale businesses.

This challenge becomes even more pronounced in tasks that require advanced reasoning, such as automated theorem proving [Wu et al., 2022], mathematical problem solving [Trinh et al., 2024], code generation [Jiang et al., 2024, Li et al., 2025a], or heuristic discovery [Romera-Paredes et al., 2024]. LLMs often fail to produce accurate responses in these scenarios in a single pass. Instead, they rely on iterative generation strategies combined with aggregation or search techniques, such as best-of-N sampling [Stiennon et al., 2020] or Monte Carlo Tree Search [Xie et al., 2024b], to refine and select the most appropriate response. Throughout this paper, we refer to these iterative strategies as Reasoning Methods.

A key limitation of current approaches lies in their static application of reasoning methods, where the same technique is applied uniformly across all user queries. However, not all reasoning methods are

equally effective or efficient for every query. This observation leads to our central research question: Could we select the most suitable reasoning method for a given user query to balance the trade-off between accuracy and efficiency *before* generating the answer?

As a starting point, we consider a universe of methods, denoted by $\mathcal{M}$. Each technique in $\mathcal{M}$ is formally characterized by a tuple $(\mathrm{LM}, \mathrm{ReStrat}, \mathrm{Agg}, \mathrm{Config}, N)$, where $\mathrm{LM}$ denotes the base language model, $\mathrm{ReStrat}$ denotes the reasoning strategy (e.g. Monte Carlo Tree Search, Beam Search, Best-of-N), $\mathrm{Config}$ is a collection of relevant configuration parameters (e.g., temperature of a sampling-based decoding strategy), $\mathrm{Agg}$ denotes an aggregation technique (e.g., majority voting or score-based voting), and $N$ is the number of candidate answers for aggregation. Importantly, this formulation is broad enough to subsume a wide variety of test-time compute methods [Snell et al., 2025], ranging from simple prompting techniques [Wei et al., 2022, Yao et al., 2023, Brown et al., 2020] and standard decoding strategies [Xie et al., 2024b, Wang et al., 2022] to more specialized intervention-based approaches [Li et al., 2023, Nguyen et al., 2025a,b]. However, in this work, we focus on a representative subset of methods rather than exhaustively covering the entire space.

**Contributions.** We introduce EPIC, an **E**nsemble **P**lann**I**ng with **C**ontrastive learning framework that recommends matching an input question and an appropriate reasoning method in the universe of methods $\mathcal{M}$. EPIC learns *jointly* the embedding of each reasoning method and a neural mapping from the input question to the embedding space. Two main components guide the learning process:

- a contrastive loss, which pulls the question embedding towards the reasoning method with the highest utility for that question. The utility value is composed of a weighted combination of the accuracy and the inference cost, measured by the number of tokens generated. The user controls the accuracy-cost trade-off through a scalar parameter, balancing the preferences across different conflicting deployment criteria.
- a regularizer term, which exploits the commonality among methods that share four components $(\mathrm{LM}, \mathrm{ReStrat}, \mathrm{Config}, \mathrm{Agg})$, but differ only by the number of candidate answers $N$. This regularizer aims to improve the sample efficiency of the training procedure by grounding these methods relatively on the scale of $N$.

At inference time, EPIC maps the test-time input question to the embedding space and selects the reasoning method with the highest similarity (or scores) for answer generation. Extensive experiments on the MATH dataset demonstrate EPIC's advantage: compared to individual reasoning models in the universe of methods, EPIC can reduce the number of tokens (or cost) by 75% while maintaining the same level of accuracy.

Our paper unfolds as follows: Section 2 discusses related work on LLM reasoning. Section 3 studies the probabilistic bounds of common aggregation methods. Section 4 delineates our EPIC framework for matching reasoning methods with input questions, and Section 5 presents the extensive numerical results of the mathematical reasoning task.

## 2   Related Work

We review advances in LLM reasoning algorithms and inference-time scaling, highlighting their emerging impact on output quality and computational efficiency.

**Reasoning algorithms and inference-time scaling.** A naive reasoning process may not generate the correct solutions for complex reasoning tasks. To identify and choose the correct solution within the distribution, Self-Consistency (SC) samples multiple outputs from the LLM and selects the final response by majority voting [Wang et al., 2022]. Another similar approach is best-of-N sampling, which uses a reward model or function to choose the answer with the highest reward [Stiennon et al., 2020]. Both methods enhance the quality of the output, but increase the computational cost by a factor of sampling times. To explore potential reasoning paths, tree-search-based methods are proposed, such as Tree-of-Thought [Yao et al., 2024], Monte Carlo Tree Search (MCTS) [Wan et al., 2024, Zhang et al., 2024, Guan et al., 2025], Forest-of-Thought [Bi et al., 2025]. Damani et al. [2025] indicates that searching over a tree structure is more effective in discovering a correct solution than simply sampling responses in parallel for more complex tasks.

Despite applying different reasoning methods to problems with various levels of complexity, inference-time scaling on these methods also significantly improves the output quality. Beeching et al. [2024]

demonstrates that the accuracy on the MATH-500 benchmark improves as the amount of test-time computation (number of generations per problem) increases for algorithms such as best-of-N, beam search, and diverse verifier tree search (DVTS). Guan et al. [2025] conduct extensive MCTS rollouts and achieve an average accuracy of 53.3% on 15 questions of AIME24 benchmark.

**Cost-effective reasoning.** Although inference-time scaling significantly enhances LLM's reasoning capabilities, this approach incurs substantial computational overhead and often leads to inefficient use of computational resources. Recent work finds that performance gains from various inference-time scaling strategies exhibit significant variability across different levels of prompt difficulty [Snell et al., 2025]. Drawing from this evidence, they effectively allocate inference-time compute according to question difficulty, with four times less computation than the best-of-N baseline. However, the method incurs considerable computational costs to assess question difficulty. Damani et al. [2025] train lightweight probes built upon LLM's hidden representations to quickly predict if allocating more computation to a question will improve the response quality. To efficiently scale best-of-N sampling, Manvi et al. [2024] introduces a highly cost-effective self-evaluation paradigm that does not rely on an external reward model, incurring costs only from generating a single token.

Whereas most studies focus on effectively and efficiently scaling a particular reasoning algorithm, we focus on pairing suitable reasoning methods with various questions, considering both accuracy and cost. We conduct our study based on OpenR [Wang et al., 2024a], an open-source framework for LLM reasoning that integrates multiple strategies, including greedy decoding, best-of-N, beam search, and MCTS.

## 3 Probabilistic Analysis of Aggregation Accuracy

We observe that many methods in the universe $\mathcal{M}$ could share common features: they could use the same base language model, reasoning strategy, configuration parameters, and aggregation methods, and they could differ by only the amount of test-time compute, or how many samples $N$ they need to generate before aggregation. To exploit this information, we first need to understand how different sample sizes $N$ affect the quality of the output. We analyze the probabilistic performance of an aggregation method for a specific question $q$ as the number of samples $N$ varies. All probability quantities in this section are conditioned on $q$, but this condition is omitted to avoid clutter. Let $\tilde{Y}$ be a random variable representing the final answer extracted from a sampled solution to a question $q$ generated by a model. Importantly, $\tilde{Y}$ refers specifically to the final answer, not the reasoning process or steps leading to it. In practice, the model is trained to enclose $\tilde{Y}$ in a LaTeX box to make extraction easier. We suppose that the support set of $\tilde{Y}$ is finite: $\mathcal{Y} = \{y_1, y_2, \ldots, y_K\}$. The stochastic generation process of a model specified by the tuple $(\mathrm{LM}, \mathrm{ReStrat}, \mathrm{Config})$ produces a probability distribution over $\mathcal{Y}$:

$$\Pr(\tilde{Y} = y_k) = p_k, \quad \text{where} \quad \sum_{k=1}^{K} p_k = 1 \text{ and } p_k \geq 0 \quad \forall k.$$

Without any loss of generality, we denote $y_1 \in \mathcal{Y}$ as the only correct answer to the question $q$. After sampling $N$ independent samples following the above distribution, an aggregation method Agg is applied to obtain the output answer. We focus on characterizing the probability that the output answer is $y_1$, which means that the output answer is a correct solution to question $q$.

### 3.1 Majority Voting

Given $N$ samples generated by the language model, majority voting counts the frequency of each unique answer among $N$ candidate answers. Then it outputs the answer with the highest count as the output answer. We refer to this aggregation method as Majority_Vote. We have the following result.

**Theorem 3.1** (Majority voting). *If $p_1 > p_k$ for all $k = 2, \ldots, K$, then*

$$\Pr(\textit{Majority\_Vote picks } y_1) \geq 1 - \sum_{k=2}^{K} e^{-N\left(\sqrt{p_1} - \sqrt{p_k}\right)^2}. \tag{1a}$$

*If $p_1 < p_k$ for some $k = 2, \ldots, K$, then*

$$\Pr(\textit{Majority\_Vote picks } y_1) \leq e^{-N\left(\sqrt{p_k} - \sqrt{p_1}\right)^2}. \tag{1b}$$

Note that the bound (1a) approaches 1 as $N \to \infty$, which implies perfect accuracy. In contrast, the bound (1b) approaches 0 as $N \to \infty$, implying a complete failure.

## 3.2 Aggregation using Summation of Scores

Given $N$ samples generated by a model, we first pass them through a reward model to obtain a reward score for each sample. Two popular types of reward models are Outcome Reward Models (ORM) [Cobbe et al., 2021, Yu et al., 2023], which provide a single scalar reward for each complete solution trajectory, and Process Reward Models (PRM) [Luo et al., 2024, Lightman et al., 2023a], which provide step-by-step feedback and aggregate it, typically by summing or taking the minimum, to obtain a final score for the sample. While PRM is used throughout this work, our method is flexible and can be applied with any reward models.

For each unique answer, we sum the PRM scores of all samples that generate that answer. The final outcome is selected as the answer with the highest total (summed) reward score across all samples [Wang et al., 2024a, Li et al., 2022]. We suppose that PRM returns a score for answer $y_k$ following a Gaussian distribution $\mathcal{N}(\mu_k, \sigma_k^2)$ for all $k$. We call this aggregation method PRM_Vote. We have the following result.

**Theorem 3.2** (Voting with score sum). *If $p_1 \mu_1 > p_k \mu_k$ for all $k = 2, \ldots, K$, then*

$$\Pr(\textit{PRM\_Vote picks } y_1) \geq 1 - \sum_{k=2}^{K} \inf_{t_k > 0} \exp\left( N p_1 \left( e^{-t_k \mu_1 + \frac{1}{2} t_k^2 \sigma_1^2} - 1 \right) + N p_k \left( e^{t_k \mu_k + \frac{1}{2} t_k^2 \sigma_k^2} - 1 \right) \right).$$
(2a)

*If $p_1 \mu_1 < p_k \mu_k$ for some $k = 2, \ldots, K$, then*

$$\Pr(\textit{PRM\_Vote picks } y_1) \leq \inf_{t > 0} \exp\left( N p_k \left( e^{-t \mu_k + \frac{1}{2} t^2 \sigma_k^2} - 1 \right) + N p_1 \left( e^{t \mu_1 + \frac{1}{2} t^2 \sigma_1^2} - 1 \right) \right). \quad (2b)$$

All infimum problems in (2) are convex optimization problems. While no analytical expression for the optimal value $t$ is available, we could tractably find $t_k$ for each term using Newton's method. Moreover, we could observe a similar conclusion as $N$ tends to infinity: the bound (2a) approaches 1 while the bound (2b) approaches 0.

## 3.3 Aggregation using Maximum of Scores

This aggregation method follows the same setup as in Section 3.2: given $N$ samples, we use the PRM to assign a reward score to each sample. For each unique answer, we take the maximum PRM score among all samples that produce that answer. The final prediction is the answer with the highest such maximum. We suppose that PRM returns a score for answer $y_k$ following a Gaussian distribution $\mathcal{N}(\mu_k, \sigma_k^2)$ for all $k$. We call this aggregation method PRM_Max. We have the following result.

**Theorem 3.3** (Voting with score maximum). *Let*

$$\Phi_k(t) := \Phi\left( \frac{t - \mu_k}{\sigma_k} \right), \quad k = 1, \ldots, K,$$

*where $\Phi$ is the cumulative distribution function of the standard normal distribution.*

*If $\sigma_1 > \sigma_k$ for all $k = 2, \ldots, K$, then*

$$\Pr(\textit{PRM\_Max picks } y_1) \geq 1 - \sum_{k=2}^{K} \inf_{t \in \mathbb{R}} \left\{ (1 - p_1 [1 - \Phi_1(t)])^N + 1 - (1 - p_k [1 - \Phi_k(t)])^N \right\}. \quad (3a)$$

*If $\sigma_k > \sigma_1$ for some $k = 2, \ldots, K$, then*

$$\Pr(\textit{PRM\_Max picks } y_1) \leq \inf_{t \in \mathbb{R}} \left\{ (1 - p_k [1 - \Phi_k(t)])^N + 1 - (1 - p_1 [1 - \Phi_1(t)])^N \right\}. \quad (3b)$$

All infimum problems in (3) are one-dimensional and can be efficiently solved using standard numerical methods. Moreover, we observe similar asymptotic behavior as $N$ increases: if $\sigma_1 > \sigma_k$ for all $k = 2, \ldots, K$, the bound in (3a) approaches 1 as $N \to \infty$, while if $\sigma_k > \sigma_1$ for some $k$, the bound in (3b) approaches 0 as $N \to \infty$.

# 4 Ensemble Planning with Contrastive Learning

Given a universe of methods $\mathcal{M} = \{1, \ldots, M\}$ consisting of $M$ reasoning methods in total, EPIC aims to create an ensemble model on $\mathcal{M}$ that assigns to any input question $x$ from the test environment an appropriate method $i \in \mathcal{M}$ that could deliver a desirable accuracy-cost trade-off. We first discuss our modeling of the accuracy-cost trade-off in Section 4.1, then we describe the training phase and inference phase in Sections 4.2 and 4.3. We conclude this section by discussing our design choices.

## 4.1 Accuracy-Cost and Utility

We possess a training dataset of $n$ question-answer pairs denoted as $\mathcal{D} = \{x_j, y_j\}_{j=1}^n$, where $x_j$ is a question statement, and $y_j$ is the corresponding true answer. In the training phase, we deploy a reasoning model $\Phi_i$, $i \in \mathcal{M}$, to each question $x_j$. The generated solution is $\Phi_i(x_j)$. We record whether $\Phi_i(x_j)$ is accurate by comparing it to the ground-truth answer $y_j$, and obtain the accuracy signal

$$a_{i,j} = \text{Accuracy}(\Phi_i(x_j), y_j) \in [0, 1]. \tag{4}$$

If $\Phi_i$ is a deterministic method, then (4) is a simple binary indicator $\text{Accuracy}(\Phi_i(x_j), y_j) = \mathbb{1}(\Phi_i(x_j) = y_j)$. When $\Phi_i$ is a stochastic method, then we average the accuracy over five seed numbers to get a percentage accuracy. The value $a_{i,j}$ indicates whether method $i$ succeeds in answering question $j$. Moreover, we also record how many tokens the method $i \in \mathcal{M}$ costs to generate the answer. This token count is denoted by $\tilde{c}_{i,j} > 0$. Because $a_{i,j} \in [0, 1]$, we normalize the token count by passing $\tilde{c}_{i,j}$ through a non-decreasing function $\phi : \mathbb{R}_+ \to \mathbb{R}_+$, then dividing by the maximum transformed cost to ensure that the cost $c_{i,j} \in [0, 1]$ has the same scale with $a_{i,j}$:

$$c_{i,j} = \frac{\phi(\tilde{c}_{i,j})}{\max_{i' \in \mathcal{M}}(\phi(\tilde{c}_{i',j}))}.$$

To balance cost and success rate, we establish the utility function that is the convex combination of accuracy $a_{i,j}$ and normalized cost $c_{i,j}$ as

$$u(a_{i,j}, c_{i,j}) = \lambda a_{i,j} + (1 - \lambda)(1 - c_{i,j}), \tag{5}$$

where $\lambda \in [0, 1]$ is a trade-off parameter, and the utility admits a value between 0 and 1. If $\lambda = 0$, then $u(a_{i,j}, c_{i,j}) = 1 - c_{i,j}$, which implies that the utility depends only on the generation cost. In this way, we tend to favor the cheapest reasoning method, regardless of how effective it is at generating accurate answers. On the other end of the spectrum, when $\lambda = 1$, $u(a_{i,j}, c_{i,j}) = a_{i,j}$, implying that the utility is entirely derived from the accuracy. In this way, we tend to favor the most powerful reasoning method, regardless of its cost. To simplify the notation, we omit the parameter $\lambda$, and use the shorthand $u_{i,j} = u(a_{i,j}, c_{i,j})$.

The product of the data preparation process is a processed dataset $\{x_j, (u_{i,j})_{i \in \mathcal{M}}\}_{j=1}^n$ containing the training question and the corresponding utility of each reasoning method for that question. This dataset will be used in the subsequent contrastive learning process.

## 4.2 Contrastive Representation Learning with Probability Regularization

We now describe the core component of our framework that matches the input question with the appropriate reasoning method. We represent each question $x_j$ in the training dataset by its features $f_j \in \mathbb{R}^D$. A lightweight neural network $g_\theta : \mathbb{R}^D \to \mathbb{R}^d$ maps each question feature vector $f_j$ to produce a dense embedding $g_\theta(f_j)$ in a $d$-dimensional vector space. EPIC aims for an information compression with $d \ll D$. Moreover, each reasoning method $i \in \mathcal{M}$ is assigned a trainable embedding vector $v_i \in \mathbb{R}^d$, which is the same dimension as the question embeddings. EPIC uses a simple multi-layer perceptron for $\theta$.

We now train the question embedding network parameter $\theta$ and the method embedding vectors $v_i$ *jointly*. One component in the training loss is the popular contrastive loss function, InfoNCE loss [Oord et al., 2018]. We identify a positive method for each question $x_j$, denoted as $m_+(x_j)$. Given the utility values defined in (5), we can identify the method with the highest utility for question $x_j$:

$$m_+(x_j) = \arg\max_{i \in \mathcal{M}} u(a_{i,j}, c_{i,j}),$$

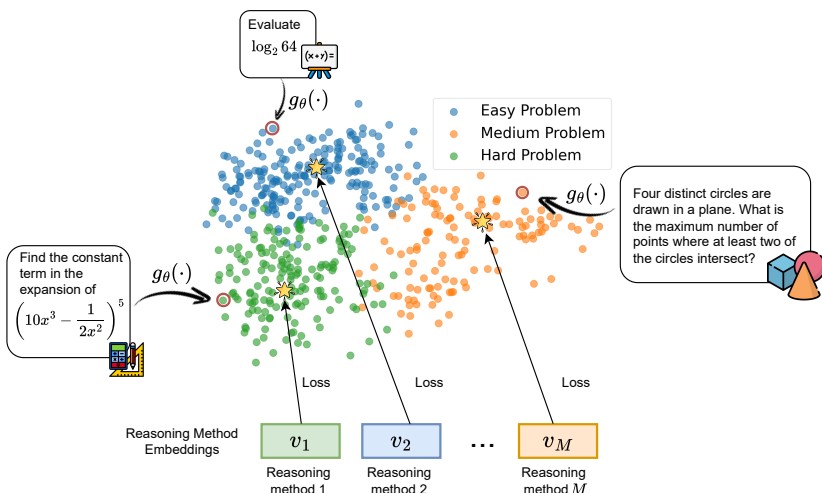

Figure 1: Our method employs the regularized representation learning loss (8) to learn both the reasoning method representation vectors, denoted as $v_1, \ldots, v_M$, and the question embedding network parameters $\theta$. During inference, we route suitable math questions to the appropriate reasoning method by computing the similarity between the input questions and the learned method representations. Color codes on problem difficulty levels are provided for illustration purposes only.

This leads to the contrastive loss component:

$$\ell_{\text{contrastive}}(\theta, v_1, \ldots, v_M) = \frac{1}{n} \sum_{j=1}^{n} - \log \left( \frac{\exp(s(g_\theta(f_j), v_{m_+(x_j)}))}{\sum_{i \in \mathcal{M}} \exp(s(g_\theta(f_j), v_i))} \right). \tag{6}$$

Above, $s$ is a similarity score function that measures the similarity of a question embedding $g_\theta(f_j)$ with the method embedding $v_i$. Standard choices for $s$ are the dot product similarity measure or the negative 2-norm. The contrastive component (6) aims to pull $g_\theta(f_j)$ close to the positive method $v_{m_+(x_j)}$, and push $g_\theta(f_j)$ far away from the negative methods $i \neq m_+(x_j)$. The loss in (6) is the categorical cross-entropy loss of classifying the positive method, with the fraction inside the logarithm being the model's prediction.

The second component of the loss function is a regularization term: Two methods that share the same tuple (base model, reasoning strategy, aggregation technique and configuration) but differ *only* by the compute budget $N$ should conform to a relative performance metric because they both inherit the same stochastic generator. We postulate the following regularization term:

$$\ell_{\text{reg}}(\theta, v_1, \ldots, v_M) = \frac{1}{n} \sum_{j=1}^{n} \sum_{\substack{(i,i') \in \mathcal{M} \\ (i,i') \text{ differ only by } N}} \left( \frac{s(g_\theta(f_j), v_i)}{s(g_\theta(f_j), v_{i'})} - \frac{\text{target}_i^j}{\text{target}_{i'}^j} \right)^2. \tag{7}$$

This regularizer promotes the fraction of the similarities to be close to the fraction of the target quantities. Ideally, we should use $\text{target}_j^i = \Pr(\text{method } i \text{ picks the correct answer for question } x_j)$, which is the intrinsic characteristic of the stochastic generator. However, this probability value is not readily available, therefore we leverage the bounds in Section 3 as target values, and empirically compute these target values as follows: For each question $j$ and core configuration (generation method, temperature, aggregation method, etc.), we generate 80 solutions (5 independent runs of $N = 16$ with different seed numbers) to obtain a set of distinct solutions $y_1, \ldots, y_K$. We then estimate the parameters $\hat{p}_k, \hat{\mu}_k, \hat{\sigma}_k$ from these 80 solutions. We can then empirically identify whether the lower or upper bound of the probability is active and assign the target value as either the lower or upper bound with the corresponding size $N$. The bound provided in Theorem 3.3 is valid when $N$ is large enough. For smaller $N$, we use the empirical accuracy as an alternative.

Combining two loss terms (6) and (7), we obtain the training problem

$$\min_{\theta} \min_{v_1, \ldots, v_M \in \mathbb{R}^d} \ell_{\text{contrastive}}(\theta, v_1, \ldots, v_M) + \tau \ell_{\text{reg}}(\theta, v_1, \ldots, v_M), \tag{8}$$

where $\tau > 0$ is a hyperparameter aiming to promote the sample efficiency of the training procedure.

### 4.3 Inference Time Matching

At inference time, we pass any new question $x_{\text{new}}$, or equivalently its feature vector $f_{\text{new}}$, through the trained network $g_\theta$ to obtain the question embedding $g_\theta(f_{\text{new}})$. We then find the top-1 reasoning method by $m^\star = \arg\max_{i \in \mathcal{M}} \ s(g_\theta(f_{\text{new}}), v_i)$, that maximizes the similarity score between the question and the trained representation vector $v_1, \ldots, v_M$ of the reasoning models. We then deploy method $m^\star$ to answer this question.

### 4.4 Discussions

We now discuss the necessity and importance of the design choices of our EPIC method.

**Discussion 1 (Questions' feature vector).** There are multiple ways to obtain the feature vector $f_j$ for each question $x_j$. For example, we can take $f_j$ as the activation of the last token of $x_j$ extracted from one of the layers (potentially the last layer) of the language model. This approach does not incur any additional memory requirement because we do not need to load any auxiliary models onto the device. However, the activation dimension of the language models is usually high: for example, in Qwen2.5-Math-7B-Instruct [Yang et al., 2024a], $D = 3584$. This high dimensionality could prohibit efficient training of the representation parameters $\theta$. Alternatively, we can use a lightweight model to map $x_j$ to $f_j$. This could incur additional memory overhead but generate a lower $D$ as input to the network $g_\theta$. In the experiment, we will use a lightweight sentence embedding `all-MiniLM-L6-v2`[1] that has only 22.6 million parameters and incurs only 80MB of additional VRAM. The corresponding feature dimension is $D = 384$.

**Discussion 2 (Importance of embedding network $g_\theta$).** Given the question features $f_1, \ldots, f_n \in \mathbb{R}^D$, one could simplify the representation learning problem (8) by optimizing the method embedding vectors $v_1, \ldots, v_M$ directly on the space of $\mathbb{R}^D$. This is equivalent to setting $d = D$, and letting $g_\theta$ collapse into an identity mapping. However, the proximity between two question features $f_j$ and $f_{j'}$ does not convey enough information about the similarity regarding hardness, resource utilization, and suitability with methods. Moreover, learning the method embedding $v_j$ on $\mathbb{R}^D$ is more difficult than on the smaller dimension space $\mathbb{R}^d$. Hence, learning in $\mathbb{R}^D$ is inefficient. This observation necessitates the use of a lightweight question map $g_\theta$.

**Discussion 3 (Adaptive method insertion).** Given a universe of models $\mathcal{M}$, problem (8) optimizes one vector $v_i$ for a reasoning model $i \in \mathcal{M}$. Alternatively, we could use another network $h_\vartheta$ that could take a (text) description of a reasoning method and output the respective embedding vector in the representation space $\mathbb{R}^d$. Having the second network $h_\vartheta$ could unlock several new capabilities: (i) for a new reasoning method that is not in $\mathcal{M}$, we could quickly obtain its embedding and predict its performance on the questions, (ii) we could inverse engineer to design a better reasoning method. Unfortunately, training $h_\vartheta$ requires a meaningful textual description of the reasoning methods. This is currently outside the scope of this paper, and we leave it for subsequent work.

## 5 Numerical Experiments

In this section, we present numerical experiments showcasing the performance of EPIC on the math answering task. Experiments for the code generation task are relegated to Appendix D.4.

**Dataset.** We use the MATH dataset [Hendrycks et al., 2021] as a training set, utilizing its training split of 7,500 math problems with solutions, as defined in Hendrycks et al. [2021]. For the code generation experiment, we use the LiveCodeBench dataset [Jain et al., 2025]. More details are in Appendix D.4. For evaluation, we test on the MATH500 test split, which contains 500 samples, as defined in Lightman et al. [2023b]. We also use the test set of the GSM8K [Cobbe et al., 2021] dataset to evaluate the transferability of the method embedding vectors learned in Section 4.2.

**Base models.** We employ Qwen2.5-Math-7B-Instruct[2] as our generation model, and math-shepherd-mistral-7b-prm [Wang et al., 2024b][3] as our reward model in the PRM framework. These models are fixed throughout our main experiments. For transferability experiments, we augment our universe of

---

[1]`https://huggingface.co/sentence-transformers/all-MiniLM-L6-v2`
[2]`https://huggingface.co/Qwen/Qwen2.5-Math-7B-Instruct`
[3]`https://huggingface.co/peiyi9979/math-shepherd-mistral-7b-rl`

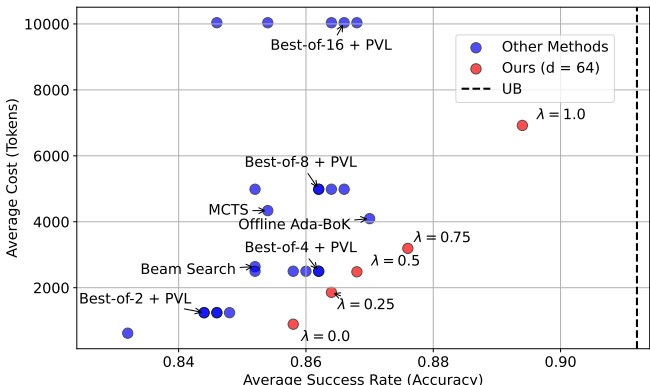

Figure 2: Average success rate and token counts on the test set with embedding dimension $d = 64$. Our ensemble planner performances with varying $\lambda \in \{0, 0.25, 0.5, 0.75, 1\}$ are highlighted in red, and individual reasoning models in $\mathcal{M}$ are plotted in blue. The boundary of the ensemble planners covers the individual models in the universe $\mathcal{M}$. The Upper Bound (UB) under $\mathcal{M}$ is the proportion of questions that at least one method in $\mathcal{M}$ could successfully solve.

methods to include Qwen2.5-Math-1.5B. For code generation experiments, we use Qwen2.5-Coder-3B-Instruct and Qwen2.5-Coder-7B-Instruct.

**Performance metrics.** We use the accuracy to measure the quality of generation and average token counts to evaluate the efficiency of each method. For accuracy, we use the automatic grading[4] provided by previous work Lightman et al. [2023b] to evaluate the accuracy of a generated solution in (4). To measure average token counts, we set the hyperparameter 'max_new_token' to 2048 for all methods and compute the average number of tokens generated.

**Universe of methods $\mathcal{M}$.** We generate $\mathcal{M}$ consisting of 81 distinct methods, spanning a variety of reasoning strategies, aggregation techniques, and parameter configurations. A complete description is provided in Appendix B.

**Dataset generation for contrastive learning.** For a deterministic method $i$ in $\mathcal{M}$, we run inference on each question $x_j$ once and record the accuracy and number of generated tokens. For the sampling method, we run the inference on each question five times to obtain a mean estimate of $a_{i,j}$ and $c_{i,j}$.

**Baselines.** We compare EPIC against three categories of baselines: (i) individual reasoning methods from the universe $\mathcal{M}$, (ii) strong large-model references including **DeepSeek-V3** and **OpenAI-o1-mini**, and (iii) alternative reasoning selection methods—**RA**, **Offline Ada-BoK** [Damani et al., 2025], **DRA-$\lambda$**, and **CL-$\lambda$**. All ensemble baselines and EPIC are trained and evaluated on the same $\mathcal{M}$ for fair comparison. Further details are provided in Appendix C.

**Reproducibility.** All experiments are conducted on a single machine with $8\times$ NVIDIA RTX A5000 GPU and Intel(R) Xeon(R) Gold 6148 CPU @ 2.40GHz.

### 5.1 Comparison between EPIC and Baselines

In the first experiment, we benchmark how EPIC, an ensemble model, outperforms individual reasoning models in the universe $\mathcal{M}$. Table 1 presents the test performance comparison between our ensemble planner and individual reasoning methods, computed based on average accuracy and the number of generated tokens. We apply the regularization parameter $\tau = 10^{-3}$ based on its better numerical results than other values shown in Appendix D. Our method with $\lambda = 0.25$ achieves an accuracy of 86.4%, matching the best-of-16 approach while using significantly fewer tokens: EPIC generates 1859.2 tokens while best-of-16 generates 10036.2 tokens. On a relative scale, this is a 5x reduction in the token counts at the same accuracy level. Compared to beam search, our approach

---

[4] https://github.com/openai/prm800k/blob/main/prm800k/grading/grader.py

at $\lambda = 0.25$ achieves better accuracy with a 29.5% reduction in token usage. With $\lambda = 1.00$, our method achieves the highest accuracy (89.4%) at a significantly lower token count (6,921.7).

To better visualize the performance of EPIC, we plot in Figure 2 a scatter plot locating the accuracy-cost trade-off of EPIC instances and representative reasoning models from $\mathcal{M}$. Our EPIC instances (red) all lie on the frontier, thereby boosting the performance of the inference phase.

Since EPIC is an ensemble method constructed from the universe of reasoning methods $\mathcal{M}$, the performance of EPIC is constrained by the capacity of the universe $\mathcal{M}$ itself. We could compute the best possible accuracy of the whole universe $\mathcal{M}$ on the test set: $\mathcal{M}$ could solve a question if there is at least one method from $\mathcal{M}$ that could generate a correct answer. Computing this value yields an upper bound of approximately 91.2%. Figure 2 highlights that our method provides a flexible trade-off between efficiency and accuracy, approaching the upper bound (dashed vertical line) while maintaining computational efficiency.

Table 1: Average accuracy and number of generated tokens on MATH500 using different reasoning methods and language models (full results in Appendix 4). Methods above the blue line are either upper bounds under $\mathcal{M}$ or are not included in $\mathcal{M}$. Methods above the green line correspond to individual single-reasoning configurations (without selection modules). Methods above the brown line do not support accuracy–cost trade-offs. We compare EPIC-$\lambda$, DRA-$\lambda$, and CL-$\lambda$ at various trade-off settings (groups separated by gray lines). Best results in each section are highlighted.

| Method | Accuracy ↑ | Average Token Count ↓ |
|---|---|---|
| OpenAI-o1-mini* [Jaech et al., 2024] | 90.0 | - |
| Deepseek-V3* [Liu et al., 2024] | 90.2 | - |
| Upper Bound under $\mathcal{M}$ | 91.2 | - |
| CoT-G | 83.2 | 620.4 |
| Best-of-2 | 84.8 | 1242.5 |
| Best-of-4 | 86.2 | 2499.4 |
| Best-of-8 | 86.6 | 4986.8 |
| Best-of-16 | 86.8 | 10036.2 |
| MCTS | 85.4 | 4338.1 |
| Beam-search | 85.2 | 2638.1 |
| RA | 84.4 | 1752.4 |
| Offline Ada-BoK | 87.0 | 4095.2 |
| DRA-0.25 | 86.2 | 2453.6 |
| CL-0.25 | 86.0 | 2275.6 |
| **EPIC-0.25** | **86.4** | **1859.2** |
| DRA-0.5 | 86.4 | 5719.3 |
| CL-0.5 | 86.6 | 5320.2 |
| **EPIC-0.5** | **86.8** | **2482.6** |
| DRA-0.75 | 86.4 | 7523.2 |
| CL-0.75 | 87.0 | 7524.6 |
| **EPIC-0.75** | **87.6** | **3192.9** |
| DRA-1.0 | 87.0 | 10542.2 |
| CL-1.0 | 87.8 | 10923.4 |
| **EPIC-1.0** | **89.4** | **6921.7** |

*Method not in $\mathcal{M}$. We obtained results from Liu et al. [2024].

## 5.2 Transferability

We now examine the transferability of EPIC across both model scales and datasets. Table 2 summarizes results for two complementary settings: (a) transferring from the MATH to the GSM8K dataset, and (b) applying EPIC in a cost-aware multi-model environment with Qwen2.5-Math-1.5B and Qwen2.5-Math-7B.

### 5.2.1 Evaluating EPIC with Cost-Aware Multi-Model Reasoning

EPIC remains effective even when reasoning methods use heterogeneous base models. Previously, our universe $\mathcal{M}$ contained 81 methods built solely on Qwen2.5-Math-7B-Instruct. We now augment this space with an additional 81 methods using Qwen2.5-Math-1.5B-Instruct. Because larger models

Table 2: EPIC performance comparison across datasets and model sizes. (a) GSM8K results where EPIC is trained on the MATH dataset using $d = 64$ and $\lambda = 0.25$. (b) Performance and cost comparison for Qwen2.5-Math-1.5B and 7B models. Best results in each column are highlighted.

(a) GSM8K results

| Method | Accuracy ↑ | Tokens ↓ |
|---|---|---|
| CoT-G | 93.5 | 297.0 |
| Best-of-2 | 94.0 | 594.6 |
| Best-of-4 | 94.0 | 1195.6 |
| Best-of-8 | 94.3 | 2412.3 |
| Best-of-16 | 94.2 | 5019.2 |
| **EPIC** | **95.0** | **2085.5** |

(b) Performance and cost with Qwen2.5 models

| Method | Accuracy ↑ | Cost ↓ |
|---|---|---|
| 1.5B-CoT-G | 76.0 | **856.5** |
| 1.5Best-of-4 | 78.6 | 3454.5 |
| 7B-CoT-G | 83.2 | 4342.8 |
| 7B-Best-of-4 | 86.2 | 17495.8 |
| 1.5Best-of-16 | 79.4 | 62663.2 |
| 7B-Best-of-16 | 86.8 | 70253.4 |
| EPIC ($\lambda = 0.25$) | 86.2 | 8047.8 |
| EPIC ($\lambda = 1.0$) | **89.0** | 35705.4 |

are computationally more expensive, we approximate the cost of each method as the product of its parameter count (in billions) and the number of generated tokens. This proxy aligns with real-world inference costs; alternatively, FLOPs or API pricing could be used.

Table 2b shows that EPIC adapts effectively across cost regimes. At $\lambda = 0.25$, EPIC achieves an accuracy of 86.2, matching the 7B-Best-of-4 method, while reducing the cost by over 50% (8047.8 vs. 17495.8). At $\lambda = 1.0$, EPIC attains the highest overall accuracy (89.0), outperforming 7B-Best-of-16 while maintaining roughly half the computational cost (35705.4 vs. 70253.4). These results demonstrate EPIC's capacity to balance accuracy and cost by dynamically leveraging reasoning methods from different model sizes.

### 5.2.2 Transfer to Another In-Domain Dataset

To further assess generalization, we evaluate EPIC trained on MATH and test it on GSM8K [Cobbe et al., 2021], another widely used arithmetic reasoning benchmark. As shown in Table 2a, GSM8K is a simpler dataset; hence, absolute gains are smaller. Nonetheless, EPIC achieves the best accuracy (95.0%) while requiring fewer tokens than high-cost baselines such as Best-of-8 or Best-of-16. This indicates that EPIC's learned representations transfer across related reasoning distributions and continue to yield efficient inference-time behavior.

### 5.3 Additional Experiments

We perform ablation studies to better understand the impact of the representation dimension, the regularization parameter $\tau$, and the tradeoff parameter $\lambda$, the utility function, the transferability, and the generalization on the code generation task. In the experiment of representation dimension, we fix $\lambda = 0.5$ and vary dimension $d \in \{16, 32, 64, 128\}$. Our results show a general trend: as $d$ increases, accuracy improves, while average token count stabilizes or slightly decreases. This result empirically confirms the expectation that increasing the embedding dimension could boost the performance of our method. In the experiment on the impact of $\lambda$, we observe a clear cost-accuracy trade-off as we fix $d = 64$ and varies $\lambda \in \{0.00, 0.25, 0.50, 0.75, 1.00\}$. In the ablation study of the utility function, we switch to an alternative functional form, as shown in equation (5), and observe a decrease in performance, indicating that our design choice is superior. Due to space constraints, further experimental details are provided in Appendix D.

## 6 Conclusion

We introduced EPIC, the **E**nsemble **P**lann**I**ng with **C**ontrastive learning framework, a contrastive learning framework that plans optimal reasoning strategies for language models by matching questions to suitable methods. Our analysis established new accuracy bounds for common aggregation techniques, which directly inform a regularization term to guide more sample-efficient learning. Experiments on mathematical reasoning benchmarks demonstrate that EPIC leverages these theoretical insights to achieve strong improvements in both accuracy and inference cost, showcasing the value of principled modeling for reasoning method selection.

**Acknowledgments.** Viet Anh Nguyen gratefully acknowledges the support from the CUHK's Improvement on Competitiveness in Hiring New Faculties Funding Scheme, UGC ECS Grant 24210924, and UGC GRF Grant 14208625.

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

# A  Proofs of Section 3

After sampling $N$ independent candidate solutions from the distribution, we aggregate them to obtain the output answer. We focus on characterizing the probability that the output answer is $y_1$, meaning that the output answer is a correct solution to question $q$.

Let $\tilde{\boldsymbol{C}}^{(N)} = (\tilde{C}_1^{(N)}, \ldots, \tilde{C}_K^{(N)})$ denote the counts of each solutions after $N$ samples, so that $\sum_{k=1}^{K} \tilde{C}_k^{(N)} = N$. The count random vector $\tilde{\boldsymbol{C}}^{(N)}$ follows the multinomial distribution:

$$\Pr(\tilde{\boldsymbol{C}}^{(N)} = \boldsymbol{c}) = \frac{N!}{c_1! \ldots c_K!} \prod_{k=1}^{K} p_k^{c_k} \tag{9}$$

for any vector $\boldsymbol{c} = (c_1, \ldots, c_K)$ of natural numbers summing up to $N$.

## A.1  Majority Vote Analysis

The majority vote selects $y_1$, the correct answer, with probability one if $c_1 > c_k$ for all $k \neq 1$, or with some probability $0 < w < 1$ if $c_1 \geq c_k$ for all $k \neq 1$ with strict equality $c_1 = c_k$ for some $k \neq 1$. The value $w$ represents the probability of choosing $y_1$ in cases of ties.

Thus, the exact probability that the majority vote selects $y_1$ is bounded by

$$\sum_{\substack{c_1 + \cdots + c_K = N \\ c_1 > c_k \ \forall k \neq 1}} \frac{N!}{c_1! \cdots c_K!} \prod_{k'=1}^{K} p_{k'}^{c_{k'}} \ \leq \ \Pr(\text{majority vote picks } y_1) \ \leq \sum_{\substack{c_1 + \cdots + c_K = N \\ c_1 \geq c_k \ \forall k \neq 1}} \frac{N!}{c_1! \cdots c_K!} \prod_{k'=1}^{K} p_{k'}^{c_{k'}}.$$

However, these bounds are computationally intractable for large $N$ due to the combinatorial explosion of possible vote count configurations. Moreover, they do not provide clear insight into how the selection probability changes as $N$ varies, limiting their practical utility for analytical understanding or approximation.

We now state the theorem that bounds the probability that the count of one solution is less than or equal to another:

**Proposition A.1** (Count upper-bound). *Assume $p_a > p_b$ for any pair of distinct indices $a, b \in \{1, \ldots, K\}$, $a \neq b$. Then, we have*

$$\Pr(\tilde{C}_a^{(N)} \leq \tilde{C}_b^{(N)}) \leq \exp\left(-N \left(\sqrt{p_a} - \sqrt{p_b}\right)^2\right). \tag{10}$$

*Proof of Proposition A.1.* For each draw $u = 1, \ldots, N$, let $C_u$ denote the selected category with $\Pr(C_u = k) = p_k$. Define

$$\tilde{C}_k^{(N)} = \sum_{u=1}^{N} \mathbb{1}_{\{C_u = k\}}, \qquad k = 1, \ldots, K.$$

Then the difference of counts between bins $a$ and $b$ can be written as

$$\tilde{C}_a^{(N)} - \tilde{C}_b^{(N)} = \sum_{u=1}^{N} X_u, \qquad X_u := \mathbb{1}_{\{C_u = a\}} - \mathbb{1}_{\{C_u = b\}}.$$

The variables $(X_u)_{u=1}^{N}$ are i.i.d., therefore for any $t > 0$, Markov's inequality implies

$$\Pr(\tilde{C}_a^{(N)} \leq \tilde{C}_b^{(N)}) = \Pr\left(e^{-t(\tilde{C}_a^{(N)} - \tilde{C}_b^{(N)})} \geq 1\right) \leq \mathbb{E}\left[e^{-t(\tilde{C}_a^{(N)} - \tilde{C}_b^{(N)})}\right] = \left(\mathbb{E}[e^{-tX_1}]\right)^N.$$

Conditioning on $C_1$ yields

$$\mathbb{E}[e^{-tX_1}] = p_a e^{-t} + p_b e^{t} + (1 - p_a - p_b).$$

Substituting this expression into the previous bound gives

$$\Pr(\tilde{C}_a^{(N)} \leq \tilde{C}_b^{(N)}) \leq \exp\left(N \log\left(p_a e^{-t} + p_b e^{t} + 1 - p_a - p_b\right)\right).$$

To optimize the bound, define $h(t) = p_a e^{-t} + p_b e^t + 1 - p_a - p_b$. Minimizing $h(t)$ over $t > 0$ leads to the first-order optimality condition

$$h'(t) = -p_a e^{-t} + p_b e^t = 0 \quad \Longrightarrow \quad e^{2t^*} = \frac{p_a}{p_b} \quad \Longrightarrow \quad t^* = \tfrac{1}{2} \log\left(\frac{p_a}{p_b}\right).$$

Since $p_a > p_b$, we have $t^* > 0$, which is valid. Plugging $t^*$ back into $h(t)$, we obtain

$$h(t^*) = 1 - p_a - p_b + 2\sqrt{p_a p_b} = 1 - (\sqrt{p_a} - \sqrt{p_b})^2.$$

Therefore, we obtain

$$\Pr(\tilde{C}_a^{(N)} \le \tilde{C}_b^{(N)}) \le \left(1 - (\sqrt{p_a} - \sqrt{p_b})^2\right)^N.$$

Finally, using $\log(1 - x) \le -x$ for $x \in (0, 1)$, we obtain the exponential form

$$\Pr(\tilde{C}_a^{(N)} \le \tilde{C}_b^{(N)}) \le \exp\left(-N(\sqrt{p_a} - \sqrt{p_b})^2\right).$$

The proof is complete. $\qquad\square$

We are now ready to prove Theorem 3.1.

*Proof of Theorem 3.1.* Recall that we assume $y_1$ is the correct solution. Consider the case where $p_1 > p_k$ for all $k = 2, \ldots, K$. We obtain by applying Proposition A.1:

$$\Pr(\text{majority vote picks } y_1) \ge \Pr\left(\bigcap_{k=2}^{K} \left\{\tilde{C}_1^{(N)} > \tilde{C}_k^{(N)}\right\}\right)$$

$$= 1 - \Pr\left(\bigcup_{k=2}^{K} \left\{\tilde{C}_1^{(N)} \le \tilde{C}_k^{(N)}\right\}\right)$$

$$\ge 1 - \sum_{k=2}^{K} \Pr\left(\tilde{C}_1^{(N)} \le \tilde{C}_k^{(N)}\right)$$

$$\ge 1 - \sum_{k=2}^{K} e^{-N\left(\sqrt{p_1} - \sqrt{p_k}\right)^2}.$$

Now consider the case where there exists $k \in \{2, \ldots, K\}$ such that $p_k > p_1$:

$$\Pr(\text{majority vote picks } y_1) \le \Pr\left(\bigcap_{k=2}^{K} \left\{\tilde{C}_1^{(N)} \ge \tilde{C}_k^{(N)}\right\}\right) \le \Pr\left(\tilde{C}_1^{(N)} \ge \tilde{C}_k^{(N)}\right)$$

$$\le e^{-N\left(\sqrt{p_k} - \sqrt{p_1}\right)^2},$$

where the last inequality follows from Proposition A.1. $\qquad\square$

### A.2 Aggregation using Summation of Scores

Let $\tilde{C}_k^{(N)}$ be the count of the solution $y_k$, and let each of the $\tilde{C}_k^{(N)}$ PRM scores be independently distributed with the same distribution $\tilde{S}_{ku} \sim \mathcal{N}(\mu_k, \sigma_k^2)$ for all $u$. Define $\tilde{U}_k^{(N)} = \sum_{u=1}^{\tilde{C}_k^{(N)}} \tilde{S}_{ku}$. We now state the theorem that bounds the probability that the total score for one solution is less than or equal to that of another.

**Theorem A.2** (Sum upper-bound). *Suppose that $p_a \mu_a > p_b \mu_b$ for some distinct $a, b \in \{1, \ldots, K\}$. Then*

$$\Pr\left(\tilde{U}_a^{(N)} \le \tilde{U}_b^{(N)}\right) \le \inf_{t>0} \exp\left(N p_a \left(e^{-t\mu_a + \frac{1}{2}t^2 \sigma_a^2} - 1\right) + N p_b \left(e^{t\mu_b + \frac{1}{2}t^2 \sigma_b^2} - 1\right)\right). \quad (11)$$

*Moreover, under this condition, the right-hand side decays exponentially in $N$.*

*Proof.* For each draw $u = 1, \ldots, N$, let $C_u$ denote the category selected, with $\Pr(C_u = k) = p_k$. Let $\tilde{S}_{ku} \sim \mathcal{N}(\mu_k, \sigma_k^2)$ be independent across both $k$ and $u$, and independent of $(C_u)_{u=1}^N$. Then

$$\tilde{U}_k^{(N)} = \sum_{u=1}^N \tilde{S}_{ku} \mathbb{1}_{\{C_u = k\}}, \qquad k = 1, \ldots, K,$$

so that

$$\tilde{U}_a^{(N)} - \tilde{U}_b^{(N)} = \sum_{u=1}^N X_u, \quad \text{where } X_u = \tilde{S}_{au} \mathbb{1}_{\{C_u = a\}} - \tilde{S}_{bu} \mathbb{1}_{\{C_u = b\}}.$$

The random variables $(X_u)_{u=1}^N$ are i.i.d. Hence, for any $t > 0$, Markov's inequality yields

$$\Pr\left(\tilde{U}_a^{(N)} \le \tilde{U}_b^{(N)}\right) \le \mathbb{E}\left[e^{-t(\tilde{U}_a^{(N)} - \tilde{U}_b^{(N)})}\right] = \left(\mathbb{E}[e^{-tX_1}]\right)^N.$$

Conditioning on $C_1$ gives

$$\mathbb{E}[e^{-tX_1}] = p_a \mathbb{E}[e^{-t\tilde{S}_{a1}}] + p_b \mathbb{E}[e^{t\tilde{S}_{b1}}] + (1 - p_a - p_b).$$

Using the moment generating function of a normal random variable gives us

$$\mathbb{E}[e^{t\tilde{S}_{k1}}] = e^{t\mu_k + \frac{1}{2}t^2 \sigma_k^2}.$$

Finally, we obtain

$$\mathbb{E}\left[e^{-t(\tilde{U}_a^{(N)} - \tilde{U}_b^{(N)})}\right] = \left(p_a e^{-t\mu_a + \frac{1}{2}t^2 \sigma_a^2} + p_b e^{t\mu_b + \frac{1}{2}t^2 \sigma_b^2} + 1 - p_a - p_b\right)^N.$$

To simplify the expression, we use $\log(1 + x) \le x$ for $x > -1$. We obtain

$$\Pr\left(\tilde{U}_a^{(N)} \le \tilde{U}_b^{(N)}\right) \le \exp\left(N p_a (e^{-t\mu_a + \frac{1}{2}t^2 \sigma_a^2} - 1) + N p_b (e^{t\mu_b + \frac{1}{2}t^2 \sigma_b^2} - 1)\right).$$

Optimizing over $t > 0$, we obtain

$$\Pr\left(\tilde{U}_a^{(N)} \le \tilde{U}_b^{(N)}\right) \le \inf_{t>0} \exp\left(N p_a (e^{-t\mu_a + \frac{1}{2}t^2 \sigma_a^2} - 1) + N p_b (e^{t\mu_b + \frac{1}{2}t^2 \sigma_b^2} - 1)\right).$$

To verify the exponential decay, we define

$$F(t) = p_a \left(e^{-t\mu_a + \frac{1}{2}t^2 \sigma_a^2} - 1\right) + p_b \left(e^{t\mu_b + \frac{1}{2}t^2 \sigma_b^2} - 1\right).$$

We have $F(0) = 0$ and

$$F'(0) = -p_a \mu_a + p_b \mu_b < 0,$$

so for small $t > 0$, $F(t) < 0$. Therefore, the exponent is negative and the bound decays exponentially in $N$. $\qquad\square$

We can now prove Theorem 3.2.

*Proof of Theorem 3.2.* Consider the case where $p_1 \mu_1 > p_k \mu_k$ for all $k = 2, \ldots, K$. Following an analogous argument as in the proof of Theorem 3.1, we have

$$\Pr(\text{PRM\_Vote picks } y_1) \ge 1 - \sum_{k=2}^K \inf_{t_k > 0} \exp\left(N p_1 \left(e^{-t_k \mu_1 + \frac{1}{2}t_k^2 \sigma_1^2} - 1\right) + N p_k \left(e^{t_k \mu_k + \frac{1}{2}t_k^2 \sigma_k^2} - 1\right)\right).$$

Consider the case where there exists $k$ such that $p_k \mu_k > p_1 \mu_1$. We have

$$\Pr(\text{PRM\_Vote picks } y_1) \le \Pr(\tilde{U}_k^{(N)} \le \tilde{U}_1^{(N)})$$
$$\le \inf_{t>0} \exp\left(N p_k \left(e^{-t\mu_k + \frac{1}{2}t^2 \sigma_k^2} - 1\right) + N p_1 \left(e^{t\mu_1 + \frac{1}{2}t^2 \sigma_1^2} - 1\right)\right).$$

This completes the proof. $\qquad\square$

### A.3 Aggregation using Maximum of Scores

To ease the readability, we recall the setup for this section. Let $\tilde{C}_k^{(N)}$ be the count of the solution $y_k$, and let each of the $\tilde{C}_k^{(N)}$ PRM scores be independently distributed with the same distribution $\tilde{S}_{ku} \sim \mathcal{N}(\mu_k, \sigma_k^2)$ for all $u$. We define $\tilde{M}_k^{(N)} = \max_{1 \leq u \leq \tilde{C}_k^{(N)}} \tilde{S}_{ku}$.

**Proposition A.3** (Max PRM upper bound). *Assume $\sigma_a > \sigma_b$ for some distinct pair $a, b \in \{1, \ldots, K\}$, $a \neq b$. Then, for any $N \geq 1$,*

$$\Pr(\tilde{M}_a^{(N)} \leq \tilde{M}_b^{(N)}) \leq \inf_{t \in \mathbb{R}} \left\{ (1 - p_a[1 - \Phi(\frac{t - \mu_a}{\sigma_a})])^N + 1 - (1 - p_b[1 - \Phi(\frac{t - \mu_b}{\sigma_b})])^N \right\}.$$

*Moreover, the right-hand side converges to zero as $N \to \infty$.*

*Proof of Proposition A.3.* We start by characterizing the distribution of $\tilde{M}_b^{(N)}$. Conditional on $\tilde{C}_b^{(N)} = m$, we have

$$\Pr(\tilde{M}_b^{(N)} \leq t \mid \tilde{C}_b^{(N)} = m) = [\Phi_b(t)]^m,$$

where $\Phi_b(t) = \Phi((t - \mu_b)/\sigma_b)$ is the CDF of $\mathcal{N}(\mu_b, \sigma_b^2)$. Substituting this into the law of total probability yields

$$\begin{aligned}
\Pr(\tilde{M}_b^{(N)} \leq t) &= \sum_{m=0}^{N} \Pr(M = m) \Pr(\tilde{M}_b^{(N)} \leq t \mid M = m) \\
&= \sum_{m=0}^{N} \binom{N}{m} p_b^m (1 - p_b)^{N-m} [\Phi_b(t)]^m \\
&= \sum_{m=0}^{N} \binom{N}{m} (p_b \Phi_b(t))^m (1 - p_b)^{N-m} \\
&= (1 - p_b + p_b \Phi_b(t))^N \\
&= (1 - p_b[1 - \Phi_b(t)])^N.
\end{aligned}$$

Taking the complement gives

$$\Pr(\tilde{M}_b^{(N)} > t) = 1 - (1 - p_b[1 - \Phi_b(t)])^N.$$

An identical argument gives

$$\Pr(\tilde{M}_a^{(N)} \leq t) = (1 - p_a[1 - \Phi_a(t)])^N, \qquad \Phi_a(t) = \Phi\left(\frac{t - \mu_a}{\sigma_a}\right).$$

Next, we have that

$$\begin{aligned}
\Pr(\tilde{M}_a^{(N)} \leq \tilde{M}_b^{(N)}) &= \Pr(\tilde{M}_a^{(N)} \leq t \text{ or } \tilde{M}_b^{(N)} > t \text{ for all } t) \\
&\leq \Pr(\tilde{M}_a^{(N)} \leq t \text{ or } \tilde{M}_b^{(N)} > t \text{ for any } t) \\
&\leq \Pr(\tilde{M}_a^{(N)} \leq t) + \Pr(\tilde{M}_b^{(N)} > t),
\end{aligned}$$

where the last inequality follows from the union bound. Hence we have the following for every $t \in \mathbb{R}$:

$$\Pr(\tilde{M}_a^{(N)} \leq \tilde{M}_b^{(N)}) \leq (1 - p_a[1 - \Phi\left(\frac{t - \mu_a}{\sigma_a}\right)])^N + 1 - (1 - p_b[1 - \Phi\left(\frac{t - \mu_b}{\sigma_b}\right)])^N$$

Since this inequality holds for all $t$, we may optimize the bound by taking the infimum over $t \in \mathbb{R}$:

$$\Pr(\tilde{M}_a^{(N)} \leq \tilde{M}_b^{(N)}) \leq \inf_{t \in \mathbb{R}} \left\{ (1 - p_a[1 - \Phi\left(\frac{t - \mu_a}{\sigma_a}\right)])^N + 1 - (1 - p_b[1 - \Phi\left(\frac{t - \mu_b}{\sigma_b}\right)])^N \right\}.$$

We now analyze the asymptotic decay of the bound as $N \to \infty$. Let $t_N = \mu_b + \sigma_b \sqrt{2(1 + \varepsilon) \log N}$ for some fixed $\varepsilon \in (0, 1)$. We define

$$z_{b,N} = \frac{t_N - \mu_b}{\sigma_b} = \sqrt{2(1 + \varepsilon) \log N}, \qquad z_{a,N} = \frac{t_N - \mu_a}{\sigma_a} = \frac{\sigma_b}{\sigma_a} \sqrt{2(1 + \varepsilon) \log N} - \frac{\mu_a - \mu_b}{\sigma_a}.$$

Then, the bound evaluated at $t_N$ becomes:

$$\Pr(\tilde{M}_a^{(N)} \leq \tilde{M}_b^{(N)}) \leq \inf_{t \in \mathbb{R}} \left\{ (1 - p_a[1 - \Phi(z_{a,N})])^N + 1 - (1 - p_b[1 - \Phi(z_{b,N})])^N \right\}.$$

We use the standard Mills inequalities, valid for all $x > 0$:

$$\frac{x}{x^2 + 1} \varphi(x) \leq 1 - \Phi(x) \leq \frac{\varphi(x)}{x}, \qquad \varphi(x) = (2\pi)^{-1/2} e^{-x^2/2}.$$

We first study $1 - (1 - p_b[1 - \Phi(z_{b,N})])^N$. For any $0 \leq x \leq 1$ it holds that $1 - (1 - x)^N \leq Nx$. We use it to have

$$1 - (1 - p_b[1 - \Phi(z_{b,N})])^N \leq Np_b[1 - \Phi(z_{b,N})].$$

Using the Mills upper bound, we obtain

$$1 - \Phi(z_{b,N}) \leq \frac{\varphi(z_{b,N})}{z_{b,N}} = \frac{1}{\sqrt{2\pi}\, z_{b,N}} N^{-(1+\varepsilon)}.$$

Hence, we have

$$Np_b[1 - \Phi(z_{b,N})] \leq \frac{p_b}{\sqrt{2\pi}\, z_{b,N}} N^{-\varepsilon} \to 0 \qquad \text{as } N \to \infty.$$

Next we study $(1 - p_a[1 - \Phi(z_{a,N})])^N$. We start with the inequality

$$0 \leq (1 - p_a[1 - \Phi(z_{a,N})])^N \leq \exp(-Np_a[1 - \Phi(z_{a,N})]),$$

which holds because $1 - x \leq e^{-x}$ for any $x \in [0,1]$.

If $z_{a,N} \leq 0$ for sufficiently large $N$, then $1 - \Phi(z_{a,N}) \geq 1/2$, so that

$$(1 - p_a[1 - \Phi(z_{a,N})])^N \leq (1 - p_a/2)^N \to 0 \quad \text{as } N \to \infty.$$

Otherwise, if $z_{a,N} > 0$, the lower Mills bound gives

$$1 - \Phi(z_{a,N}) \geq \frac{C}{z_{a,N}} e^{-z_{a,N}^2/2}$$

for some constant $C > 0$. Hence,

$$Np_a[1 - \Phi(z_{a,N})] \geq C' \frac{N}{z_{a,N}} e^{-z_{a,N}^2/2}.$$

Using the definition of $z_{a,N}$, we obtain

$$z_{a,N}^2 = \frac{\sigma_b^2}{\sigma_a^2} 2(1 + \varepsilon) \log N + o(\log N).$$

Substituting into the inequality yields us

$$Np_a[1 - \Phi(z_{a,N})] \geq C'' N^{1 - (1+\varepsilon)\sigma_b^2/\sigma_a^2} e^{o(\log N)}.$$

for some constant $C'' > 0$. Since $\sigma_a > \sigma_b$, we can choose $\varepsilon > 0$ sufficiently small so that

$$1 - (1 + \varepsilon)\frac{\sigma_b^2}{\sigma_a^2} > 0,$$

implying that $Np_a[1 - \Phi(z_{a,N})] \to \infty$, and hence

$$(1 - p_a[1 - \Phi(z_{a,N})])^N \to 0.$$

Combining with the analysis for $b$, we finally obtain

$$\Pr(\tilde{M}_a^{(N)} \leq \tilde{M}_b^{(N)}) \to 0 \quad (N \to \infty).$$

This completes the proof. $\qquad\qquad\square$

We are now ready to prove Theorem 3.3.

*Proof of Theorem 3.3.* Consider the case where $\sigma_1 > \sigma_k$ for all $k = 2, \ldots, K$. Following an analogous argument as in the proof of Theorem 3.1, we use Proposition A.3 to have

$$\Pr(\text{PRM\_Max picks } y_1) \geq 1 - \sum_{k=2}^{K} \inf_{t \in \mathbb{R}} \left\{ (1 - p_1[1 - \Phi(\frac{t - \mu_1}{\sigma_1})])^N \right.$$
$$\left. + 1 - (1 - p_k[1 - \Phi(\frac{t - \mu_k}{\sigma_k})])^N \right\},$$

where $\Phi$ is the cumulative distribution function of the standard normal distribution. By Proposition A.3, each term in the summation tends to zero as $N \to \infty$. Therefore the probability that PRM\_Max correctly selects $y_1$ tends to 1.

In the alternative case, suppose there exists some $k$ such that $\sigma_k > \sigma_1$ Following an analogous argument as in the proof of Theorem 3.1, we use Proposition A.3 to have

$$\Pr(\text{PRM\_Max picks } y_1) \leq \inf_{t \in \mathbb{R}} \left\{ (1 - p_k[1 - \Phi(\frac{t - \mu_k}{\sigma_k})])^N \right.$$
$$\left. + 1 - (1 - p_1[1 - \Phi(\frac{t - \mu_1}{\sigma_1})])^N \right\}.$$

Furthermore, by Proposition A.3, the bound tends to 0 when $N \to \infty$.

This completes the proof. □

## B Universe of Methods

In this section, we provide a systematic description of our universe of methods $\mathcal{M}$ as mentioned in Section 5. Each method is represented by a tuple $(\text{LM}, \text{ReStrat}, \text{Agg}, \text{Conf}, N)$, where each component is defined as follows.

**Language Model** ($\text{LM}$). For our main experiments, we consider a single base model: `Qwen2.5-Math-7B-Instruct`. In Appendix 5.2.1, we extend our pool of methods by additionally using `Qwen2.5-Math-1.5B-Instruct`.

**Reasoning Strategies** ($\text{ReStrat}$). We consider four primary groups of reasoning methods:

- **Greedy Search:** The deterministic strategy denoted as COT-G that always selects the highest-probability next token.

- **Best-of-$N$:** The model samples $N$ complete responses for each question according to the output token distribution, controlled by a temperature hyperparameter. The temperature adjusts the randomness of sampling: lower temperatures make the output more deterministic, while higher temperatures increase diversity. If $N = 1$, this produces a single, randomly sampled response. When $N > 1$, aggregation strategies (described below) are applied to select a final answer from the $N$ candidates.

- **Beam Search:** The model first generates $N$ distinct first steps, each evaluated by a PRM. The top $N/m$ steps with the highest PRM scores are kept, where $N/m \in \mathbb{Z}$. For each retained first step, the model generates $m$ second steps, forming $N$ partial solutions. This process repeats until $N$ complete solutions are produced or we reach the maximum number of beam expansions (50 in our case).

- **Monte Carlo Tree Search:** This method, denoted as MCTS, formulates response generation as a tree search problem, iteratively exploring possible answers by balancing exploration and exploitation. At each step, MCTS selects the most promising node based on a selection policy (e.g., Upper Confidence Bound), expands new response candidates using the base model, evaluates them through rollouts or verifier models, and back-propagates the scores to refine future selections. The search continues until we obtain $N$ complete solutions.

**Configuration** ($\mathrm{Conf}$). The configuration presents all the hyperparameters of a reasoning strategy. While almost all of them are fixed as default values suggested by the language model report and repository OpenR [Wang et al., 2024a], we vary the temperature of decoding in the set $\{0.4, 0.7, 1.0\}$ to explore different levels of diversity.

**Aggregation methods** ($\mathrm{Agg}$). For reasoning strategies that produce multiple candidate responses, we employ aggregation methods to select the final answer. These methods include:

- Majority_Vote (`MV`): Select the most frequently occurring answer from multiple samples.

- PRM_Vote_Min (`PVM`): For each generation, use the PRM to score each step, select the minimum score within the generation, and ultimately choose the generation with the highest *sum of* minimum score across all samples.

- PRM_Vote_Last (`PVL`): For each generation, use the PRM to score each step, select the score associated with generating the last step, and ultimately choose the generation with the highest *sum of* scores across all samples

- PRM_Max_Min (`PMM`): For each generation, use the PRM to score each step, select the minimum score within the generation, and ultimately choose the generation with the highest minimum score across all samples.

- PRM_Max_Last (`PML`): For each generation, use the PRM to score each step, select the score associated with generating the last step, and ultimately choose the generation with the highest score across all samples.

Notably, except for Majority_Vote, other methods are required to call the reward model.

**Candidate solution size** ($N$). We vary the number of candidate solutions in $N \in \{1, 2, 4, 8, 16\}$.

Overall, the universe $\mathcal{M}$ consists of 81 reasoning models that span different reasoning strategies and configurations. The detailed composition of the universe $\mathcal{M}$ is summarized in Table 3. Specifically, we construct $\mathcal{M}$ by combining reasoning strategies, decoding settings, and aggregation methods. The full enumeration yields 81 unique configurations: 60 Best-of-$N$ variants, 10 Beam Search variants, 10 MCTS variants, and one deterministic CoT-G configuration.

Table 3: Construction of the universe $\mathcal{M}$ (81 methods) by strategy, search budget $N$, decoding temperature temp, and aggregation $\mathrm{Agg} \in \{\mathtt{MV}, \mathtt{PVM}, \mathtt{PVL}, \mathtt{PMM}, \mathtt{PML}\}$.

| Strategy | $N$ values | temp values | #Agg choices | Count |
|---|---|---|---|---|
| Best-of-$N$ | $\{2, 4, 8, 16\}$ | $\{0.4, 0.7, 1.0\}$ | 5 | $4 \times 3 \times 5 = 60$ |
| Beam Search | $\{2, 4\}$ | 0.5 | 5 | $2 \times 5 = 10$ |
| MCTS | $\{2, 4\}$ | 0.5 | 5 | $2 \times 5 = 10$ |
| CoT-G (greedy) | 1 | (greedy) | N/A | 1 |
| | | | **Total** | **81** |

## C   Baseline Descriptions and Complete Experimental Results

We present an additional ablation study focusing on another critical component of our framework: the design of the reasoning selection module. In particular, we replace our contrastive-learning and two-tower-based reasoning selection model with four baselines. We first consider two baselines that do not support an accuracy-cost trade-off:

- **Offline Ada-BoK** adapts the approach from Damani et al. [2025], which originally operates on batches of questions and manages resources at the batch level. For fair comparison, we use their 'Offline allocation' strategy, modified to operate at the individual question level without requiring access to the entire batch at test time. Here, each allocation corresponds to choosing a specific reasoning configuration.

- **Random Allocation (RA)**: At inference time, each question randomly selects a reasoning method from the available reasoning configurations.

The next two baselines, along with our method, support an explicit accuracy-cost trade-off, controlled by the hyperparameter $\lambda$, which we vary over $\{0.0, 0.25, 0.5, 0.75, 1.0\}$.

- **Multi-class classifier (CL-$\lambda$).** In this classifier-based version, the best reasoning method for each question is still determined using the utility-driven labeling (based on the parameter $\lambda$), exactly as in our original approach. However, instead of using contrastive loss with a two-tower embedding structure, we adopt a two-layer classifier placed on top of the pretrained sentence transformer network and train with a standard cross-entropy loss, commonly used in classification scenarios. We denote this classification-based ablation as CL-$\lambda$.

- **Distributional Random Allocation (DRA-$\lambda$):** Parameter $\lambda$ matches the trade-off parameter used in our approach. During training, each question is labeled with the reasoning method having the highest $\lambda$-adjusted score prediction. At inference, reasoning methods are randomly drawn from the observed training distribution.

Table 4: Average accuracy and the number of generated tokens on MATH500 for different methods and models. Methods above the blue line are either Upper Bound under $\mathcal{M}$ or not in $\mathcal{M}$. Methods above the green line are individual single-reasoning configurations (no selection module involved). Methods above the brown line do not support accuracy-cost trade-off. Below, we compare EPIC-$\lambda$, DRA-$\lambda$, and CL-$\lambda$ at different trade-off settings (groups separated by gray lines). Best results in each section are in bold.

| Base Model | Method | Accuracy $\uparrow$ | Average Token Count $\downarrow$ |
|---|---|---|---|
| Qwen2.5 72B Base | CoT* [Yang et al., 2024b] | 80.0 | - |
| QwQ 32B | CoT* [Yang et al., 2024b] | 83.2 | - |
| OpenAI-o1-mini | CoT* [Jaech et al., 2024] | 90.0 | - |
| Deepseek-V3 | CoT* [Liu et al., 2024] | 90.2 | - |
| Qwen2.5-Math-7B-Instruct | Upper Bound under $\mathcal{M}$ | 91.2 | - |
| Qwen2.5-Math-7B-Instruct | CoT-G | 83.2 | 620.4 |
| Qwen2.5-Math-7B-Instruct | Best-of-2 | 84.8 | 1242.5 |
| Qwen2.5-Math-7B-Instruct | Best-of-4 | 86.2 | 2499.4 |
| Qwen2.5-Math-7B-Instruct | Best-of-8 | 86.6 | 4986.8 |
| Qwen2.5-Math-7B-Instruct | Best-of-16 | 86.8 | 10036.2 |
| Qwen2.5-Math-7B-Instruct | MCTS | 85.4 | 4338.1 |
| Qwen2.5-Math-7B-Instruct | Beam-search | 85.2 | 2638.1 |
| Qwen2.5-Math-7B-Instruct | RA | 84.4 | 1752.4 |
| Qwen2.5-Math-7B-Instruct | Offline Ada-BoK | 87.0 | 4095.2 |
| Qwen2.5-Math-7B-Instruct | DRA-0.0 | 85.6 | 1248.4 |
| Qwen2.5-Math-7B-Instruct | CL-0.0 | 85.2 | **606.7** |
| Qwen2.5-Math-7B-Instruct | EPIC-0.0 | **85.8** | 892.9 |
| Qwen2.5-Math-7B-Instruct | DRA-0.25 | 86.2 | 2453.6 |
| Qwen2.5-Math-7B-Instruct | CL-0.25 | 86.0 | 2275.6 |
| Qwen2.5-Math-7B-Instruct | EPIC-0.25 | **86.4** | **1859.2** |
| Qwen2.5-Math-7B-Instruct | DRA-0.5 | 86.4 | 5719.3 |
| Qwen2.5-Math-7B-Instruct | CL-0.5 | 86.6 | 5320.2 |
| Qwen2.5-Math-7B-Instruct | EPIC-0.5 | **86.8** | **2482.6** |
| Qwen2.5-Math-7B-Instruct | DRA-0.75 | 86.4 | 7523.2 |
| Qwen2.5-Math-7B-Instruct | CL-0.75 | 87.0 | 7524.6 |
| Qwen2.5-Math-7B-Instruct | EPIC-0.75 | **87.6** | **3192.9** |
| Qwen2.5-Math-7B-Instruct | DRA-1.0 | 87.0 | 10542.2 |
| Qwen2.5-Math-7B-Instruct | CL-1.0 | 87.8 | 10923.4 |
| Qwen2.5-Math-7B-Instruct | **EPIC-1.0** | **89.4** | 6921.7 |

*Method not in $\mathcal{M}$.

We observe from Table 4 that our EPIC-$\lambda$ strongly outperforms the simpler classification version (CL-$\lambda$) and DRA-$\lambda$ at almost every level of the parameter $\lambda$. Still, the CL-$\lambda$ method remains superior to individual single reasoning configurations, confirming that our labeling strategy based on the proposed utility function is indeed effective. Moreover, our original contrastive-learning approach with a two-tower embedding structure significantly enhances scalability: introducing new reasoning methods simply involves adding a new embedding vector without retraining the entire selection

module. This two-tower model and contrastive loss combination have proven highly advantageous over classification-based methods, both in terms of scalability and overall predictive performance.

# D  Ablation Studies and Additional Experiments

This section presents additional ablation studies to investigate the impact of various design choices in our framework. First, we examine the effect of representation embeddings and the cost-accuracy trade-off parameter $\lambda$. Second, we assess the robustness of our approach by substituting the utility function (5) with an alternative formulation. Third, we evaluate the transferability of our framework across different language models and other in-domain datasets. Finally, we present additional experiments for the code generation task.

## D.1  The Impact of the Representation Dimension

In this experiment, we study the impact of the dimension of the representation space $d$ on the performance of EPIC. One could expect that larger dimensions $d$ will give a higher representation power and thus EPIC could perform better. For simplicity, we conduct the experiments only for $\lambda = 0.25$. The average test accuracy and token counts are reported in Table 5. We could identify a global trend that, as $d$ increases, the accuracy increases, while the average token count tends to go flat or decrease. This result empirically confirms the expectation that increasing the embedding dimension could increase the ensemble's performance.

Table 5: Impact of $d$ on test accuracy and average token counts with $\lambda = 0.25$.

| $d$ | 16 | 32 | 64 | 128 |
|---|---|---|---|---|
| Accuracy ↑ | 85.6 | 85.4 | 86.4 | 86.2 |
| Average token counts ↓ | 1828.3 | 2271.4 | 1859.2 | 2004.5 |

## D.2  The Impact of the Trade-off Parameter $\lambda$

Table 6 illustrates the impact of $\lambda$ on accuracy and the number of generated tokens. As $\lambda$ increases from 0.00 to 1.00, we observe a consistent rise in accuracy from 85.8% to 89.4% and in average token counts from 892.9 to 6921.7. This trend indicates a clear cost-accuracy trade-off and can be visualized in Figure 2, where we can identify an upward trend of the red circles.

Table 6: Impact of $\lambda$ on test accuracy and average token counts with embedding dimension $d = 64$.

| $\lambda$ | 0.00 | 0.25 | 0.50 | 0.75 | 1.00 |
|---|---|---|---|---|---|
| Accuracy ↑ | 85.8 | 86.4 | 86.8 | 87.6 | 89.4 |
| Average token counts ↓ | 892.9 | 1859.2 | 2482.6 | 3192.9 | 6921.7 |

## D.3  Ablation on Utility Function

We previously presented two ablation analyses on the embedding dimensionality in Section D.1 and the trade-off parameter $\lambda$ in Section D.2. To further justify our choice of utility function, we conduct an additional ablation study using an alternative utility formulation:

$$u(a_{i,j}, c_{i,j}) = a_{i,j}^{\lambda} \times (1 - c_{i,j})^{1-\lambda},$$

where $a_{i,j}$ denotes accuracy and $c_{i,j}$ denotes cost for the $j$-th method on the $i$-th instance. We vary the trade-off parameter $\lambda$ over the set $\{0.0, 0.25, 0.5, 0.75, 1.0\}$.

Notably, for $\lambda = 0.0$ and $\lambda = 1.0$, this alternative function reduces to our original utility formulation. Therefore, we focus our comparison on the intermediate values $\lambda \in \{0.25, 0.5, 0.75\}$. For a better presentation, we denote our framework with this alternative utility function as PMU (Power Mean Utility).

As shown in Table 7, EPIC consistently outperforms the DRA baseline across all evaluated PMU configurations, demonstrating its robustness under different accuracy-cost trade-offs.

Table 7: Comparison of PMU and EPIC methods at different $\lambda$ settings. Best results in each section are in bold.

| Method | Accuracy ↑ | Average Token Count ↓ |
|---|---|---|
| PMU-0.25 | 86.0 | 2334.1 |
| EPIC-0.25 | **86.4** | **1859.2** |
| PMU-0.5 | 86.4 | 3035.4 |
| EPIC-0.5 | **86.8** | **2482.6** |
| PMU-0.75 | 87.2 | 4724.5 |
| EPIC-0.75 | **87.6** | **3192.9** |

## D.4    Results on Code Benchmark

To assess the generality of our proposed method beyond the math domain, we evaluate it on Live-CodeBench [Jain et al., 2025]. The universe of methods includes both the Chain of Thought - Greedy (CoT-G) and Best-of-$N$ sampling strategies. In the Best-of-$N$ approach, the base model generates $N$ candidate responses per question, from which the best is selected, with $N \in \{2, 4, 8, 16\}$ and the decoding temperature chosen from $0.2, 0.6$. CoT-G produces step-by-step solutions using greedy decoding (temperature set to 0). For evaluation, we adopt the pass@$k$ metric, as described in Li et al. [2025b], and test two LLM base models of differing capacities: Qwen2.5-Coder-3B-Instruct and Qwen2.5-Coder-7B-Instruct [Yang et al., 2024b]. In the code benchmark, we do not consider aggregation methods, so the regularization parameter $\tau$ is set to 0 in this experiment.

Table 8 reports the accuracy (pass@$k$) and average token counts for each method. We observe that our method, EPIC, achieves competitive or superior accuracy to all baselines at both $\lambda = 0.25$ and $\lambda = 1.0$. For the 7B model, EPIC ($\lambda = 1.0$) achieves the highest overall accuracy (61.88%), outperforming Best-of-16, while also consuming fewer tokens. Similarly, on the 3B model, EPIC ($\lambda = 1.0$) achieves the best accuracy (48.01%), exceeding the Best-of-16 baseline.

**Trade-off Control.** EPIC with $\lambda = 0.25$ achieves balanced performance, providing better accuracy than CoT-G and Best-of-2, but at a modest computational cost, highlighting the framework's flexibility in managing the trade-off of accuracy and cost.

**Efficiency at Lower Cost.** Notably, CoT-G remains the most computationally efficient method, but at the expense of lower accuracy. EPIC offers a favorable balance, substantially improving accuracy while keeping generation costs well below those of aggressive sampling strategies like Best-of-16.

Table 8: Performance comparison of reasoning methods on Qwen2.5-Coder-3B-Instruct and Qwen2.5-Coder-7B-Instruct. The best value in each column is in bold.

| Method | Qwen2.5-Coder-3B-Instruct | | Qwen2.5-Coder-7B-Instruct | |
|---|---|---|---|---|
| | Accuracy ↑ | Avg. Token Count ↓ | Accuracy ↑ | Avg. Token Count ↓ |
| CoT - Greedy | 24.85 | **580.80** | 35.81 | **505.15** |
| Best-of-2 (with best temperature) | 27.40 | 1144.73 | 41.68 | 1000.27 |
| Best-of-4 (with best temperature) | 32.88 | 2299.65 | 48.53 | 2011.99 |
| Best-of-8 (with best temperature) | 40.90 | 4672.83 | 53.82 | 4031.65 |
| Best-of-16 (with best temperature) | 46.38 | 9323.22 | 58.71 | 8034.65 |
| EPIC $\lambda = 0.25$ | 30.12 | 1025.43 | 43.44 | 813.42 |
| EPIC $\lambda = 1.0$ | **48.01** | 8349.12 | **61.88** | 7013.54 |

## E    Qualitative Results

### E.1    A Specific Reasoning Method Favors in Certain Questions

This appendix provides a curated example in Table 9 for qualitative analysis. The goal is to present a case where some individual methods in the universe $\mathcal{M}$ fail, but EPIC still produces accurate answers by selecting the most suitable one.

## E.2 Visualization and Analysis of Learned Embedding Space

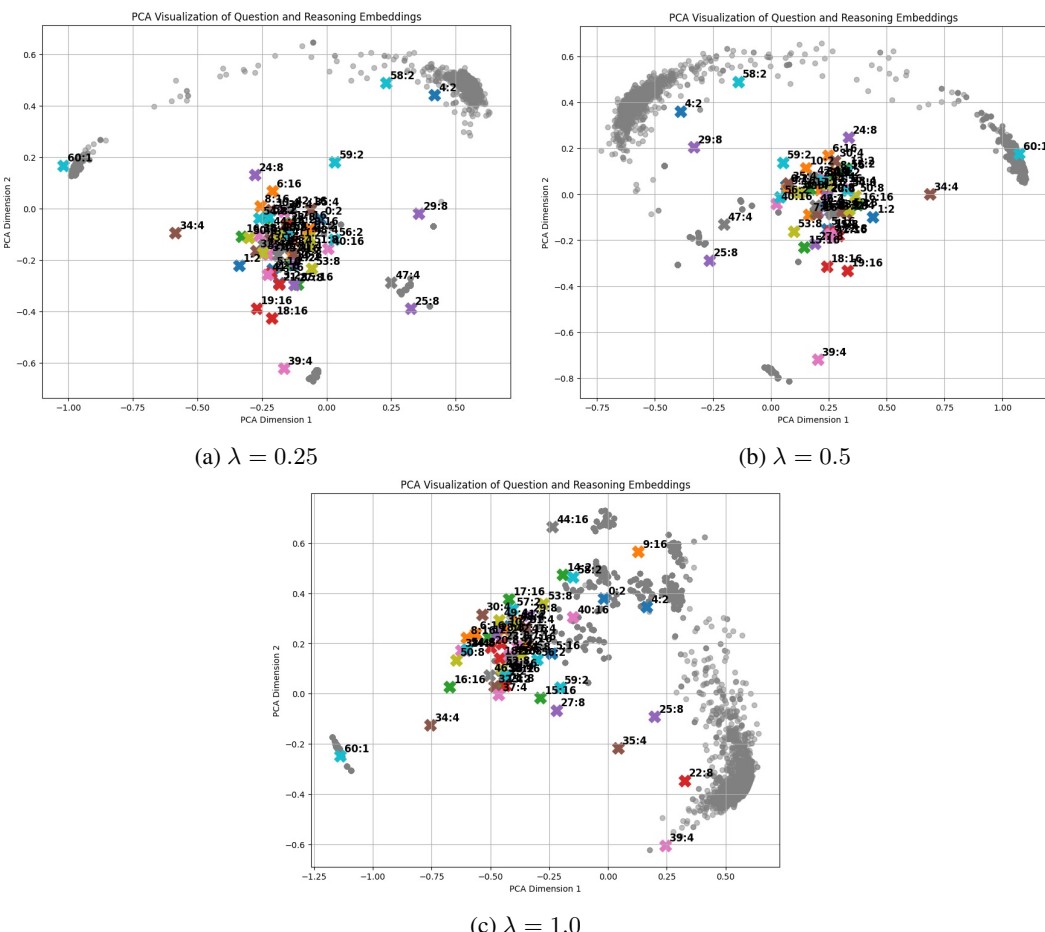

(a) $\lambda = 0.25$

(b) $\lambda = 0.5$

(c) $\lambda = 1.0$

Figure 3: PCA visualization of question (grey) and reasoning method (colored crosses) embeddings for three different settings of the utility trade-off $\lambda$. Each method is labeled by `index:num`, where `num` is the number of generated answers.

To further understand how EPIC organizes and exploits the structure of mathematical reasoning methods and questions, we visualize the learned embedding space using Principal Component Analysis (PCA). Figure 3 presents three PCA plots of the question and method embeddings for different utility trade-off values: $\lambda = 0.25$, $\lambda = 0.5$, and $\lambda = 1.0$. In these plots, the grey dots represent the embedded math questions, while each colored cross denotes a reasoning method. Each method is annotated with the format `index:num`, where `index` is the method identifier and `num` is the number of generated answers for that method.

Plot 3 illustrate how the utility trade-off parameter $\lambda$ shapes the structure of the learned embedding space:

When $\lambda = 0.25$ (cost prioritized), method embeddings are widely scattered and tend to avoid regions dense with question embeddings. Only low-cost methods are positioned near clusters of questions.

When $\lambda = 0.5$ (equal weight to accuracy and cost), higher-cost but occasionally effective methods move closer to question clusters. This reflects a balanced trade-off, where moderately accurate and moderately costly methods are preferred.

When $\lambda = 1$ (accuracy prioritized), most methods form distinct clusters around their optimal question types. This demonstrates EPIC's ability to match each question with the most suitable method.

These PCA visualizations confirm that EPIC organizes method and question representations according to the chosen trade-off. This structured alignment helps explain EPIC's strong performance discussed earlier in the paper.

**Reference Q&A**
Q: Evaluate
$$\sin(\arcsin 0.4 + \arcsin 0.5) \cdot \sin(\arcsin 0.5 - \arcsin 0.4)$$
A: From the angle addition and subtraction formulas,
$$\sin(x + y) = \sin x \cos y + \cos x \sin y, \quad \sin(x - y) = \sin x \cos y - \cos x \sin y,$$
we get $\sin(x + y)\sin(x - y) = (\sin x \cos y + \cos x \sin y)(\sin x \cos y - \cos x \sin y) = \sin^2 x - \sin^2 y$. For $x = \arcsin 0.5$ and $y = \arcsin 0.4$,
$$\sin(\arcsin 0.5 + \arcsin 0.4) \cdot \sin(\arcsin 0.5 - \arcsin 0.4) = 0.5^2 - 0.4^2 = 0.09 = \boxed{\frac{9}{100}}.$$

---

**Best-of-16 MV**
To evaluate $\sin(\arcsin 0.4 + \arcsin 0.5) \cdot \sin(\arcsin 0.5 - \arcsin 0.4)$, let $\theta = \arcsin 0.4$ and $\phi = \arcsin 0.5$, so $\sin \theta = 0.4$ and $\sin \phi = 0.5$. We need to find $\sin(\theta + \phi) \cdot \sin(\phi - \theta)$.
Using the sine addition and subtraction formulas:
$$\sin(\theta + \phi) = \sin \theta \cos \phi + \cos \theta \sin \phi, \quad \sin(\phi - \theta) = \sin \phi \cos \theta - \cos \theta \sin \phi,$$
we get:
$$\sin(\theta + \phi) \cdot \sin(\phi - \theta) = (\sin \theta \cos \phi)^2 - (\cos \theta \sin \phi)^2.$$
Now, using $\cos^2 \theta = 1 - \sin^2 \theta = 0.84$ and $\cos^2 \phi = 1 - \sin^2 \phi = 0.75$, the expression becomes:
$$(0.4 \cdot \sqrt{0.75})^2 - (\sqrt{0.84} \cdot 0.5)^2 = 0.16 \cdot 0.75 - 0.25 \cdot 0.84 = 0.12 - 0.21 = -0.09.$$
Thus, the value of the expression is:
$$\boxed{-0.09}.$$

---

**EPIC selects Best-of-16 PVL**
To evaluate $\sin(\arcsin 0.4 + \arcsin 0.5) \cdot \sin(\arcsin 0.5 - \arcsin 0.4)$, let $\alpha = \arcsin 0.4$ and $\beta = \arcsin 0.5$, so $\sin \alpha = 0.4$ and $\sin \beta = 0.5$. The expression becomes $\sin(\alpha + \beta) \cdot \sin(\beta - \alpha)$.
Using the product-to-sum identity for sines:
$$\sin x \cdot \sin y = \frac{1}{2}[\cos(x - y) - \cos(x + y)],$$
we apply it to $x = \alpha + \beta$ and $y = \beta - \alpha$, yielding:
$$\sin(\alpha + \beta) \cdot \sin(\beta - \alpha) = \frac{1}{2}[\cos 2\alpha - \cos 2\beta].$$
Using the double-angle identity $\cos 2\theta = 1 - 2\sin^2 \theta$, we get: $\cos 2\alpha = 1 - 2\sin^2 \alpha$ and $\cos 2\beta = 1 - 2\sin^2 \beta$. Substituting these, the expression becomes:
$$\sin(\alpha + \beta) \cdot \sin(\beta - \alpha) = \frac{1}{2}\left[(1 - 2\sin^2 \alpha) - (1 - 2\sin^2 \beta)\right].$$
Simplifying:
$$= \frac{1}{2} \cdot 2(\sin^2 \beta - \sin^2 \alpha) = \sin^2 \beta - \sin^2 \alpha.$$
Substituting $\sin \alpha = 0.4$ and $\sin \beta = 0.5$: $\sin^2 \beta - \sin^2 \alpha = (0.5)^2 - (0.4)^2 = 0.25 - 0.16 = 0.09$. Therefore, the value of the expression is: $\boxed{0.09}$.

Table 9: A curated sample from the MATH benchmark. This is an example where an individual method fails, but EPIC could deliver an accurate answer by selecting the most suitable one.

