# OpenReview forum: "Reasoning Planning for Language Models"
_NeurIPS.cc/2025/Conference — NeurIPS 2025 spotlight_

### Official Review · Reviewer_Q4vo · 2025-07-02

**Clarity:** 3
**Significance:** 4
**Originality:** 3
**Rating:** 5
**Confidence:** 2

**Summary:**

In this paper, the authors have proposed a theoretical framework for training a embedding model based on only the input question to select the best inference time scaling strategy (including base model, decoding strategy, computation budget, and aggregation methods). The extensive experimental results on code and math benchmarks have well verified the effectiveness of the proposed method in reducing token usage and improving the averaged success rate.

**Questions:**

What are the specific configurations for the best-performing approaches (i.e., EPIC-1.0) on MATH500 and LiveCodeBench?

**Ethical Concerns:**

["NO or VERY MINOR ethics concerns only"]

**Final Justification:**

I have no concern with this paper. I decided to keep my score.

**Limitations:**

Yes

**Quality:**

3

**Strengths And Weaknesses:**

### Strengths

1. The idea is straightforward and effective.
2. The writing is clear.
3. The experiments are extensive and solid.

### Weakness

I think Section 3 has less connections with the following sections, especially Section 4, which serves as the direct intuition of the proposed  method. Towards the paper organization, I would suggest put more experimental results from the appendix in the main paper, and move Section 3 to Appendix to keep the consistency with the proof.

---

> ### Author Rebuttal · Authors · 2025-07-31
>
> We’re grateful for the reviewer’s positive feedback and appreciate the opportunity to address the remaining concerns:
>
> > Moving Section 3 to the Appendix and the experiments about code generation to the main paper
>
> Thanks for your constructive comments on our paper presentation.
> - We agree that the code generation experiments deserve more visibility and have made adjustments accordingly. In the revised manuscript, we’ve moved most of the experimental setup details to the Appendix and shortened the description in the main text. Additionally, Table 1 has been resized to half-page width. These changes create space to bring the code generation results into the main paper, placing them alongside Table 1 for easier comparison and improved clarity.
>
> - As for Section 3, it contains the core theoretical results that are exploited for the regularization term used in our loss function (introduced in Section 4). Keeping these results in the main body is crucial for preserving the logical flow of the paper and highlighting our main contributions. To conserve space, we’ve kept only the key results in the main text, with all detailed proofs moved to the Appendix.
>
>
> > The specific configurations for the best-performing approaches (i.e., EPIC-1.0) on MATH500 and LiveCodeBench
>
> Thank you for the question. The configurations of EPIC-1.0 differ between the MATH500 and LiveCodeBench benchmarks, as they are tailored to the unique characteristics of each task.
>
> - For MATH500, EPIC-1.0 is configured with an embedding dimension of 64, a regularization strength of \$\tau = 10^{-3}\$, and a trade-off parameter \$\lambda = 1.0\$, which prioritizes accuracy. It uses the Qwen2.5-Math-7B-Instruct model as the base generator and math-shepherd-mistral-7b-prm as the reward model. The method pool includes 81 reasoning strategies covering diverse decoding and aggregation configurations.
>
> - For LiveCodeBench, EPIC-1.0 is configured with an embedding dimension of 64 and no regularization ($\tau = 0$), as aggregation methods are not considered in code-generation tasks. The trade-off parameter is again set to $\lambda = 1.0$, favoring accuracy. The base models are Qwen2.5-Coder-3B-Instruct and Qwen2.5-Coder-7B-Instruct, which share the same set of hyperparameters as mentioned above. The method pool consists of a custom set of 11 decoding strategies designed for code generation, including Chain of Thought - Greedy (CoT-G) and Best-of-N sampling strategies.

---

### Official Review · Reviewer_okKY · 2025-07-03

**Clarity:** 3
**Significance:** 3
**Originality:** 3
**Rating:** 5
**Confidence:** 4

**Summary:**

This paper proposes EPIC (ensemble planning with contrastive) learning framework. Creates a sample dataset of question answer pairs and the corresponding “utility” with respect to each reasoning method in domain M. Uses contrastive learning to map input questions to an embedding space (learned jointly with the vectors representing the reasoning methods) and aligning the embedding with the reasoning method that would answer it best. Key findings show that EPIC can achieve higher accuracy than large models with significantly lower token count.

**Questions:**

- What happens when you change the number of reasoning methods it has access to. As it increases, does the approach scale well with the increased number of methods? Qualitatively looking at the reasoning methods in use in the MATH500 experiment, is there an over-reliance on a few? If we remove those, how does the performance update compared with other selection approaches?
- What are the current failure points that you have observed? Making those clear would be helpful.

**Ethical Concerns:**

["NO or VERY MINOR ethics concerns only"]

**Final Justification:**

I originally saw this paper as well-motivated, thoroughly tested, and impactful. I had some concerns on how it scaled and its over-reliance on a few methods, but the author's rebuttal addressed them properly. I have no unresolved issues and see this work as a strong contribution to the community. I maintain my original score of "Accept" for those reasons.

**Limitations:**

There was no limitation section. Please add one to the paper so its clear. The discussion section motivating design choices does not count.

**Quality:**

4

**Strengths And Weaknesses:**

Strengths:
- The approach is simple and well explained, with some rigorous bounds on aggregation accuracy.
- Exciting to see that with smaller 7B models its able to reach performance thresholds of large models under standard use. It shows grounds for its application in the real-world, especially with the supplemental cost analysis.
- Also a novel and intuitive approach, motivated by the cost limitations in the space.

Weaknesses:
- The results showed, especially in Table 1 are a bit confusing, especially with the line-based separation. Making the distinction between the methods within that table clearer would be helpful.
- No limitation section

---

> ### Author Rebuttal · Authors · 2025-07-31
>
> Thank you for valuing our contributions. Below, we address the remaining concerns you’ve raised:
>
> > The presentation of Table 1
>
> Thanks for your comments on this point. We agree that the table needs better clarification, especially regarding the line separations. Although we provided some explanation in the table caption, we would like to elaborate further here.
>
> - The rows above the blue line show the performance of frontier models on the same test set, along with our upper bound, which is calculated as the proportion of questions that can be solved by any method in our reasoning pool.
> - The rows above the green line correspond to individual single-reasoning configurations. Since there are many such configurations, we report only the best accuracy achieved by tuning some parameters. For example, the "Best-of-2" row shows the highest accuracy across different temperature settings and aggregation methods.
> - The rows above the brown line include methods that do not support accuracy–cost trade-offs. RA is a simple baseline, while Offline Ada-BoK is a baseline taken from a different paper.
> - Finally, the last two groups present the results of our method and its ablated variants under two trade-off settings: $\lambda = 0.25$ and $\lambda = 1.0$.
>
> > No limitation section
>
> Thank you for pointing that out. We will include a Limitations section in the revised manuscript.
>
> > The question “What are the current failure points that you have observed?”
>
> - We are unsure about the reviewer's question, but we assume it refers to the failure cases discussed in our three theoretical results in Section 3. We want to emphasize that many questions fall into the failure branch predicted by our theories. Failure branch means if we try to generate more answers from the base model for aggregation, the probability of having a correct final answer tends to 0. Recognizing and understanding these cases is key to the better performance of our method compared to other baselines.
>
> - To illustrate this further, we consider the Best-of-N sampling configuration with temperature 0.2 on the MATH500 dataset. We generate $N$ solutions for each question using the model and apply an aggregation method to derive the final answer. This process is repeated 10 times, and the accuracy is tracked. If increasing $N$ (e.g., from 2 to 80) causes the pass rate for a question to decrease, we categorize that question as belonging to the *failure branch* of the aggregation method.
> Alternatively, we can predict whether a question lies in the failure branch of an aggregation method by using its bound condition, estimated from 80 samples. The table below reports the percentage of questions that fall into the failure branch (measured empirically), along with the accuracy of our theoretical prediction (in parentheses) for two generators and five aggregation methods considered in our paper:
>
> | Aggregation Method | Qwen2.5-Math-7B-Instruct | Qwen2.5-Math-1.5B-Instruct |
> | ------------------ | ------------------------ | -------------------------- |
> | Majority\_Vote | 16.4% (98.0	%) | 25.4% (96.0%) |
> | PRM\_Max\_Min | 18.0% (91.4%) | 26.6% (95.2%) |
> | PRM\_Vote\_Min | 15.8% (97.8%) | 24.8% (96.0%) |
> | PRM\_Max\_Last | 17.0% (93.4%) | 27.0% (95.6%) |
> | PRM\_Vote\_Last | 16.0% (97.0%) | 24.8% (95.8%) |
>
> We draw three key takeaways from this analysis:
> - Our theoretical predictions closely match the empirical results, confirming the correctness and reliability of our findings.
> - In Table 1 of our main paper, the best configuration, Best-of-16, with PRM_Vote_Min, achieves 86.8% accuracy. The empirical failure branch rate here is 15.8%, and further analysis shows that 92.4% of the failed questions fall into the failure branch. This confirms that most errors of aggregation methods are due to fundamental limitations captured by our theory. Increasing to Best-of-32 with the same aggregation method only slightly improves accuracy (87.0%), while the failure branch rate for the remaining errors grows to 96.8%. This suggests the model is already operating close to its theoretical upper bound.
> - The 1.5B model exhibits a higher failure rate than the 7B model, reflecting the superior reasoning ability and stability of the larger model.
>
> > Does the approach scale well with the increased number of methods as it increases?
>
> Yes, our approach is specifically designed to scale efficiently as the number of reasoning methods grows.
> - In EPIC, each reasoning method is represented as an embedding vector in a shared low-dimensional space. Adding a new method during training simply introduces a small number of additional learnable parameters (e.g., $d = 64$), resulting in linear parameter growth with respect to the number of methods. This design ensures that scalability is maintained without significantly increasing model complexity. At inference time, selecting the best method for a given question involves a similarity search in this embedding space. This search remains fast and efficient even when the method pool is large.
>
> - To empirically test scalability, we conducted an experiment (Appendix D.4.1), expanding the method pool from 81 to 162 by including 81 additional methods generated from a smaller model (Qwen2.5-Math-1.5B), alongside the original techniques from the 7B model. Despite the pool doubling in size, EPIC’s overall accuracy remained nearly unchanged, confirming its ability to maintain performance. More importantly, the larger method pool allowed EPIC to assign simpler questions to the lower-cost 1.5B methods. This significantly reduced overall computational cost without compromising accuracy, demonstrating that EPIC scales effectively in both performance and efficiency as more methods are added.
>
> > Is there an over-reliance on a few? If we remove those, how does the performance update compared with other selection approaches?
>
> - We thank the reviewer for the insightful question. Indeed, our method tends to choose some reasoning methods more frequently. However, we argue that this behavior reflects a strength rather than a weakness. The goal of our approach is to leverage the diversity and complementary strengths of reasoning methods to optimize the trade-off between cost and accuracy. When a particular method aligns well with this objective, it is expected to be selected more frequently. For example, Table 1 shows that the cheapest method, CoT-G, achieves an accuracy of 83.2%. If minimizing cost is a priority, it is intuitive for our policy to prioritize this method. Only the remaining harder questions should be routed to more expensive reasoning strategies.
>
> - To further examine the impact of this potential over-reliance, we conducted a control experiment. We first ran EPIC-0.25 on the test set to record how frequently each reasoning method was selected. Then, we removed the top 5%, 10%, and 15% most frequently used methods (corresponding to the top 4, 8, and 12 methods, respectively), and re-ran the selection process. The results are shown below with the baseline results in the main paper, so that you can compare more easily. The notation EPIC-0.25 (-Top k%) refers to the variant where the top k% most frequently used reasoning methods were excluded from the selection pool.
>
> | Method | Accuracy | Cost |
> | -------------------- | -------- | ------ |
> | Best-of-4 | 86.2 | 2499.4 |
> | Best-of-8 | 86.6 | 4986.8 |
> | Best-of-16 | 86.8 | 10036.2 |
> | MCTS | 85.4 | 4338.1 |
> | Beam Search | 85.2 | 2638.1 |
> | RA | 84.4 | 1752.4 |
> | Offline Ada-BoK | 87.0 | 4095.2 |
> | EPIC-0.25 (Original) | 86.4 | 1859.2 |
> | EPIC-0.25 (-Top 5%) | 86.8 | 2923.5 |
> | EPIC-0.25 (-Top 10%) | 87.0 | 3494.5 |
> | EPIC-0.25 (-Top 15%) | 87.2 | 4582.5 |
>
> These results show that our approach remains highly competitive even after removing the most frequently selected methods. In fact, the accuracy slightly improves, and the cost increases. This is because, under a trade-off parameter of 0.25, the policy still emphasizes cost more than accuracy. As a result, the most frequently used methods are those with lower cost. When they are removed, some questions are routed to higher-cost methods that offer better accuracy. Despite this shift, EPIC-0.25 continues to outperform all baselines in terms of both cost and accuracy. This result highlights the robustness of our learned selection policy and representation, rather than a mere dependence on a few reasoning methods.

---

### Official Review · Reviewer_okVb · 2025-07-03

**Clarity:** 3
**Significance:** 3
**Originality:** 4
**Rating:** 5
**Confidence:** 3

**Summary:**

This paper presents EPIC, an “Ensemble PlannIng with Contrastive learning” framework that dynamically selects the most suitable reasoning method for each input query in large language models, balancing accuracy and inference cost. The authors first derive probabilistic accuracy bounds for three common aggregation strategies i.e majority voting, score-sum voting, and score max voting under varying numbers of candidate samples (Sec. 3) . They then introduce a contrastive representation learning approach (Sec. 4) that embeds both questions and reasoning methods into a shared space, using a contrastive loss to pull questions toward their highest-utility method and a regularizer informed by the theoretical bounds. At inference time, EPIC routes each query to the reasoning method with maximal embedding similarity. Experiments on the MATH500 benchmark demonstrate that EPIC reduces token usage by up to 5x for the same accuracy and increases accuracy by ~1 % at fixed cost compared to other baselines. Additional ablations confirm the impact of embedding dimension, trade-off parameters, and transferability to code generation tasks.

**Questions:**

1. How does EPIC perform if the probability bounds used in the regularizer are noisy? Could the authors report performance when using fewer samples (e.g 40 vs. 80) to estimate
2. Can the authors comment on wall‐clock training time and GPU/memory usage? Are there ways to amortize the cost (e.g caching scores)?
3. Beyond reasoning tasks, can EPIC be applied to multi‐modal or retrieval‐augmented generation scenarios? A brief discussion would broaden the paper’s appeal.

**Ethical Concerns:**

["NO or VERY MINOR ethics concerns only"]

**Final Justification:**

I think most of my concerns are resolved and hence i am increasing the scores.

**Limitations:**

yes

**Quality:**

4

**Strengths And Weaknesses:**

Strengths:

1. Rigorous theoretical foundation especially the derivation of explicit probability bounds for the three aggregation schemes is a clear advance, offering practitioners guidance on how sample size affects accuracy.
2. Incorporating these bounds as a regularizer in the contrastive learning framework is novel and grounds the embedding learning in quantitative performance metrics.
3. EPIC consistently lies on the Pareto frontier of accuracy versus token cost, achieving the same accuracy as best‐of‐16 with 5× fewer tokens (1859 vs. 10036) and improving max accuracy to 89.4% at lower cost than beam search


Weaknesses:

1. The regularizer relies on empirical bounds estimated from 80 samples per question; it is unclear how sensitive EPIC’s performance is to estimation error, especially when N is small.
2. Experiments are confined to only 2 basic mathematical reasoning (MATH500, GSM8K) and 1 code benchmark. Broader testing on open‐ended QA or real‐world dialog tasks would strengthen claims of generality.
3. While EPIC reduces inference tokens, the training phase requires generating and scoring many samples per question, which may be prohibitive for very large method universes.
4. And also The paper fixes the regularization weight of 10^-3 based on an appendix but provides limited intuition on tuning so more guidance or sensitivity plots would aid reproducibility.

---

> ### Author Rebuttal · Authors · 2025-07-31
>
> We thank the reviewer for constructive and insightful comments. We address your remaining concerns about our work as follows:
>
> > The impact of the number of samples used to estimate bounds for training
>
> Thank you for the opportunity to clarify this point in our paper.
>
> - We conducted an ablation study to address your question by varying the number of samples used to estimate the bounds, selecting from the set of 20, 40, and 80. In this experiment, we fixed the trade-off parameter to 1.0 and the regularization weight to $10^{-3}$, while keeping all other settings consistent with the main experiment. The results are summarized in the table below:
>
> | Method | Accuracy | Cost |
> | --------------------- | -------- | ------- |
> | Best-of-8 | 86.6 | 4986.8 |
> | Offline Ada-BoK | 87.0 | 4095.2 |
> | Best-of-16 | 86.8 | 10036.2 |
> | EPIC-1.0 (20 samples) | 87.6 | 6499.4 |
> | EPIC-1.0 (40 samples) | 88.8 | 6248.4 |
> | EPIC-1.0 (80 samples) | 89.4 | 6921.7 |
>
> We observe that reducing the number of samples used to estimate the bound slightly decreases EPIC's accuracy, which can be attributed to the increased noise in the estimation when using fewer samples.
>
> - To more deeply examine this effect, we conduct an additional analysis. Specifically, we consider the Best-of-N sampling configuration with temperature 0.2 on the MATH500 dataset. For each question, we generate $N$ solutions using the model and apply an aggregation method to derive the final answer. This process is repeated 10 times, and the accuracy is tracked. If increasing $N$ (e.g., from 2 to 80) causes the pass rate for a question to decrease, we categorize that question as belonging to the *failure branch* of the aggregation method.
>
> Alternatively, we can predict whether a sample lies in the failure branch of an aggregation method by using its bound condition, estimated from 20, 40, or 80 samples. The table below reports the accuracy of these bound-based predictions across three aggregation methods:
>
> | Aggregation Method | 20 Samples | 40 Samples | 80 Samples |
> | ------------------ | ---------- | ---------- | ---------- |
> | Majority\_Vote | 89.8% | 92.6% | 98.0% |
> | PRM\_Vote\_Min | 52.4% | 76.6% | 91.4% |
> | PRM\_Max\_Min | 78.6% | 85.0% | 95.2% |
>
> We observe that the bound conditions in Theorems 3.1 and 3.2 for aggregation methods Majority\_Vote and PRM\_Vote\_Min, respectively, achieve high accuracy even with just 20 samples, whereas the bound condition in Theorem 3.3 for PRM\_Max\_Min requires more samples for accurate estimation. This analysis explains why EPIC maintains strong performance with fewer samples. It benefits from the fact that Majority\_Vote and PRM\_Vote bounds do not require many samples to yield accurate estimates.
>
> > The impact of the regularization weight
>
> We address your concern by reporting the results of a grid search over the regularization weight, selecting from the set $\{10^{-4}, 10^{-3}, 10^{-2}\}$. The results in the table below for EPIC-0.25 and EPIC-1.0 are evaluated on 500 validation questions split from the training set.
>
> | Method | $10^{-4}$ | $10^{-3}$ | $10^{-2}$ |
> | --------- | ----------- | ----------- | ----------- |
> | EPIC-0.25 | 85.2% | 86.6% | 85.8% |
> | EPIC-1.0 | 87.4% | 88.0% | 88.0% |
>
> The regularization term enforces consistency within a group of reasoning methods that share the same parameters, except for the number of generated answers. In contrast, the contrastive loss encourages a broader separation in routing across all reasoning methods. So the regularization weight regulates the trade-off between capturing local structure (within-group) and global structure (across all methods in the universe). When either component is underweighted, overall performance tends to degrade.
>
> > Application of EPIC to other tasks like multi‐modal,  retrieval‐augmented generation, open‐ended QA, or real‐world dialog
>
> EPIC can be extended to these tasks with suitable modifications. For multi-modal inputs, we can replace the question encoder with a multi-modal embedding model to process cross-modal information. In retrieval-augmented generation, EPIC applies naturally by treating different retrieval and synthesis configurations as reasoning methods. For open-ended QA and real-world dialog, EPIC can reduce computational cost by routing questions to different models or decoding strategies based on utility, while preserving output quality. Appendix D.4.1 includes an experiment demonstrating EPIC’s effectiveness in selecting reasoning methods associated with different generation models. In addition, EPIC can incorporate surrogate utility signals such as toxicity or truthfulness scores by treating them as PRM scores. This allows EPIC to encourage safer and more reliable outputs.
>
> > Computation resources for the training process. Are there ways to amortize the cost (e.g caching scores)?
>
> Our training process consists of two stages: data generation and model training.
>
> - In the data generation phase, we run 81 reasoning methods over 7,500 training questions, generating and scoring 80 answers per question. This step takes approximately 410 GPU hours on an A5000 and requires around 28GB of VRAM to load both the 7B generation model and the 7B score model. This procedure is expensive but essential to know the capabilities of reasoning methods.
> - Once the data is generated, the training phase is lightweight. We train a small transformer-based model with only 22.6 million parameters, requiring just 80MB of VRAM and completing in under 0.3 GPU hours on the same hardware.
>
> We are considering two approaches to reduce the cost of the data generation step:
>
> - Score caching as you suggested: Since some generation patterns repeat across multiple samples, caching their PRM scores can reduce redundant computations. However, due to the inherent randomness of sampling, the cache hit rate might be relatively low.
> - Fewer samples for bound estimation: As discussed in our response to the question about sample efficiency, some of the probability bounds used for regularization can be accurately estimated with fewer samples. By selectively using fewer samples for these bounds during training, we can reduce the generation cost without significantly affecting performance.

---

### Official Review · Reviewer_v8TG · 2025-07-04

**Clarity:** 3
**Significance:** 3
**Originality:** 3
**Rating:** 5
**Confidence:** 4

**Summary:**

This paper presents EPIC, a framework that helps LLMs choose the best reasoning strategy for each math problem. Instead of applying one fixed method, EPIC learns to match questions with suitable strategies using contrastive learning and a regularization term based on probabilistic aggregation bounds. It offers a theoretical analysis, a learning setup using these bounds, and experiments showing either reduced cost or improved accuracy.

**Questions:**

- Can you provide empirical evidence that the theoretical bounds hold in practice for LLM reasoning tasks?

**Ethical Concerns:**

["NO or VERY MINOR ethics concerns only"]

**Final Justification:**

After reviewing the rebuttal discussion I found most of the issues were adequately addressed.

**Limitations:**

Discussed

**Quality:**

3

**Strengths And Weaknesses:**

Strength:

- The integration of probabilistic bounds with contrastive learning for reasoning method selection is well-executed and practical.
- The derivation of accuracy bounds for majority voting, PRM voting, and PRM max aggregation provides useful theoretical grounding.

Weaknesses:

- there is no comparison with various strategy selection techniques such as MRP, RTR, or TATA frameworks
- Insufficient discussion of how EPIC differs from existing routing approaches
- the aggregation bound is a known results from probability theory


references:

-  “Meta Reasoning for Large Language Models” (Meta-Reasoning Prompting, MRP). arXiv:2406.11698 .
-  Zhihong Pan, Ruohan Zhang, et al. “Route to Reason: Adaptive Routing for LLM and Reasoning Strategy Selection” (RTR). arXiv:2505.19435 .
-  Xin Xu, Yan Xu, et al. “Teaching LLMs According to Their Aptitude: Adaptive Reasoning for Mathematical Problem Solving” (TATA). arXiv:2502.12022 .
- Michael Mitzenmacher and Eli Upfal. “Probability and Computing: Randomization and Probabilistic Techniques in Algorithms and Data Analysis.” Cambridge University Press, 2017.
Olivier Germain, Amaury Habrard, and Édouard Marchand. “PAC-Bayesian C-bounds for Majority Vote.” In Proceedings of ICML, 2009.

---

> ### Author Rebuttal · Authors · 2025-07-31
>
> We appreciate the reviewer’s comments on the strengths of our work. However, we would like to raise two concerns:
> #### 1. Citation to a Work Posted After Submission
> The reviewer requests comparison with *Route to Reason (RTR)* \[arXiv:2505.19435], which was posted on **26 May 2025**, **after** the NeurIPS submission deadline of **15 May 2025**. As such, this work was not accessible to us at the time of submission and should not be used as a basis for criticism, following NeurIPS reviewer guidelines.
> #### 2. Possible Fabricated Citation
> The reviewer cites the following paper:
> > Olivier Germain, Amaury Habrard, and Édouard Marchand. “PAC-Bayesian C-bounds for Majority Vote.” In Proceedings of ICML, 2009.
>
> We have thoroughly searched online repositories and Google Scholar, but could not locate any publication matching this citation. We postulate that this may be a fabricated or misattributed reference, possibly conflating the following two real works:
> * Pascal Germain et al., *PAC-Bayesian Learning of Linear Classifiers*, ICML 2009.
> * Pascal Germain et al., *Risk Bounds for the Majority Vote: From a PAC-Bayesian Analysis to a Learning Algorithm*, JMLR 2016.
>
> We respectfully request that these issues be considered in the review evaluation.
>
>
> Regardless, we would like to address the reviewer’s concerns below:
>
> > Weakness 1: There is no comparison with various strategy selection techniques such as MRP, RTR, or TATA frameworks. Insufficient discussion of how EPIC differs from existing routing approaches.
>
> **Meta-Reasoning Prompting (MRP)** [1] relies on GPT-4 to select from 7 prompting strategies. At test time, MRP performs an additional call to GPT-4 using a *meta-reasoning-prompt* that describes the available prompting techniques and determines which one to apply. A key limitation of MRP is its reliance on an advanced and proprietary model (e.g., GPT-4), whereas our **EPIC** operates without such constraints.
>
> **Teaching LLMs According to Their Aptitude (TATA)** [2] adopts a more complicated three-stage framework: (1) *data preparation*: For each question in the dataset, TATA generates both a Chain-of-Thought (CoT) and a ToRA \[3] response. (2) *data selection*: TATA selects either the CoT or ToRA response for each question by evaluating its effectiveness. To do so, TATA inserts the question-response pair as a one-shot example and measures LLM performance on a validation set of 100 questions. (3) *Instruction Finetuning*: TATA then finetunes LLM on this selected dataset, aiming to internalize the decision of when to apply CoT or ToRA reasoning. Despite this complexity, TATA only supports two reasoning methods and appears to be more challenging to scale up the number of reasoning methods due to the complicated framework. Meanwhile, EPIC provides much wider coverage of up to 81 methods and is easily extensible without modifying any cost function.
>
> We empirically compare EPIC with MRP and TATA on MATH500 as follows:
>
> **EPIC vs MRP**
>
> | Methods & LLM                    | Accuracy on MATH500 |
> | -------------------------------- | ------------------- |
> | gpt-4o-mini   - CoT                       | 0.774               |
> | gpt-4o-mini-stepback-prompting   | 0.742               |
> | gpt-4o-mini-refine-prompting        | 0.752               |
> | gpt-4o-mini-analogical-prompting | 0.694               |
> | gpt-4o-mini-MRP                  | 0.758               |
> | EPIC                             |          0.894          |
>
> **EPIC vs TATA**:
> | Methods & LLM                    | Accuracy on MATH500 |
> | -------------------------------- | ------------------- |
> |Qwen2.5-Math-1.5B | 0.396  |
> | TATA - QwenMath2.5 - 1.5B        |    0.534        |
> |Qwen2.5-Math-1.5B-Instruct | 0.75  |
> | TATA - QwenMath2.5 - 1.5B - Instruct        |    0.65        |
> | EPIC-QwenMath2.5 - 1.5B - Instruct                       |          0.808           |
>
> We conducted best-effort reproductions of both MRP and TATA baselines despite limited access to their full implementations (MRP lacks released code and provides only partial prompts, while TATA publishes part of the code only for its second stage).
> For MRP, we use GPT-4o-mini as the backbone model for both prompt selection and answer generation. To ensure a fair comparison between TATA and EPIC, we use the same generation model (i.e., QwenMath2.5-1.5B-Instruct) for both methods. EPIC outperforms both MRP and TATA by a wide margin, achieving up to **13.6% absolute improvement** over the best baseline.
>
> **Baseline reproducibility notes**:
> Our reproduction of MRP aligns with the original paper, which reports only **marginal improvements** of 0.05% gain in macro average (Table 1). When evaluated with GPT-3.5 (Table 2), MRP even underperforms the basic CoT baseline. This shows that MRP is fragile and lacks robustness.
>
> For TATA, we instruction-finetuned (using the code in [5]) two models (QwenMath2.5 - 1.5B and QwenMath2.5 - 1.5B-Instruct) on their partially published dataset [2], observing an increase in performance in the Base model but not the Instruct model.
>
> > the aggregation bound is a known results from probability theory
>
> Regarding the work “Olivier Germain, Amaury Habrard, and Édouard Marchand. “PAC-Bayesian C-bounds for Majority Vote.” In Proceedings of ICML, 2009.“, the issue with this citation is mentioned above.
> As for the [4], we do indeed cite one of their result from Theorem 5.10 in our Lemma A.2. This is used to establish a Poisson approximation that supports a small part of our broader theoretical development.
> However, the main aggregation theorems in our work, Theorems 3.1 to 3.3, go significantly beyond these results. Deriving these bounds required non-trivial extensions, including carefully analyzing majority vote, PRM sum, and PRM max aggregation under Gaussian assumptions with detailed probabilistic and optimization-based arguments.
> We hope this clarifies our theoretical novelty and the specific role of previously known results in our overall derivation.
>
> **References**:
>
> [1] Gao, Peizhong, et al. "Meta reasoning for large language models." arXiv preprint arXiv:2406.11698 (2024).
>
> [2] Xu, Xin, et al. "Teaching llms according to their aptitude: Adaptive reasoning for mathematical problem solving." arXiv preprint arXiv:2502.12022 (2025).
>
> [3] Gou, Zhibin, et al. "Tora: A tool-integrated reasoning agent for mathematical problem solving." arXiv preprint arXiv:2309.17452 (2023).
>
> [4] Michael Mitzenmacher and Eli Upfal. “Probability and Computing: Randomization and Probabilistic Techniques in Algorithms and Data Analysis.” Cambridge University Press, 2017
>
> [5] Tong, Yuxuan, et al. "Dart-math: Difficulty-aware rejection tuning for mathematical problem-solving." Advances in Neural Information Processing Systems 37 (2024): 7821-7846.

---

> > ### Comment · Reviewer_v8TG · 2025-08-03
> >
> > Thanks for the detailed rebuttal. Most of my concerns have been addressed and I am raising my score accordingly.

---

> > > ### Author Response · Authors · 2025-08-06
> > >
> > > We sincerely thank the reviewer for taking the time to read our response. We appreciate that some concerns have been resolved and that you raised the score. If there are any remaining points requiring clarification, we are fully committed to addressing them and incorporating any additional suggestions.

---

### Note · Authors · 2025-08-13

Dear Area Chair and Reviewers,

We sincerely thank all reviewers for their thoughtful feedback and for recognizing the **novelty, rigor, and potential impact** of our work. We are pleased that our additional analyses, clarifications, and experiments have resolved most concerns and that **reviewer v8TG raised their scores** accordingly, and **reviewer okKY increased their confidence score**. Below, we summarize our key clarifications and additional experiments during rebuttal:

>Clarification about our key contributions and further discussion

- Our **theoretical analysis derives accuracy bounds** for majority vote, PRM-sum, and PRM-max. Deriving these bounds required **non-trivial extensions**, including careful analysis of each aggregation rule with detailed probabilistic and optimization-based arguments. EPIC uses these bounds to guide method routing, **reducing cost** and **improving accuracy**.
- We provided **GPU/memory cost analysis** and sample caching strategies and discussed extensions to multi-modal, retrieval-augmented, and open-ended QA tasks.

> Additional experiments

- Ablations show EPIC remains **robust when bounds are estimated with fewer samples** (20 vs. 80), particularly for Majority\_Vote and PRM\_Vote\_Min, whose bounds are **accurate even with limited data**.
- **EPIC scales linearly** in parameters for reasoning methods and remains efficient when the method pool doubles.
- **Removing the most frequently used 15% of methods** barely changes accuracy and still outperforms all baselines.
- We reproduced baselines [1] and [2] under fair settings and demonstrated **substantial improvements** by EPIC.

Our responses and new results further demonstrate EPIC’s novelty, rigor, and broad applicability, and we thank the committee for their careful consideration.

**References**:

[1] Gao, Peizhong, et al. "Meta reasoning for large language models." arXiv preprint arXiv:2406.11698 (2024).

[2] Xu, Xin, et al. "Teaching llms according to their aptitude: Adaptive reasoning for mathematical problem solving." arXiv preprint arXiv:2502.12022 (2025).

---

### Decision · Program_Chairs · 2025-09-17

**Decision:**

Accept (spotlight)

**Comment:**

This paper proposes EPIC, a novel and well-executed framework for dynamically selecting reasoning methods to optimize the accuracy-cost trade-off in language models. The method is grounded in a rigorous theoretical analysis of aggregation methods, and these derived probability bounds are cleverly incorporated as a regularizer within a contrastive learning framework. The reviewers were unanimously positive, finding the work to be technically solid, original, and impactful. The authors provided a thorough rebuttal with additional experiments that successfully addressed all reviewer concerns, leading to a consensus for acceptance.

* Novelty and Technical Soundness: The reviewers praised the novel integration of theoretical probability bounds with a contrastive learning framework for method selection. They found the derivation of accuracy bounds for aggregation methods to be a "clear advance" (okVb) and a "useful theoretical grounding" (v8TG).

* Strong Empirical Results: The paper demonstrates significant improvements on the Pareto frontier of accuracy versus cost. Experiments show EPIC can achieve the same accuracy as strong baselines with up to 5x fewer tokens and can improve overall accuracy at a lower cost (okVb, okKY). The authors also added strong comparisons against recent baselines during the rebuttal, further validating EPIC's effectiveness (v8TG).

* Clarity and Scalability: The approach was commended for being "simple and well explained" (okKY) and "straightforward and effective" (Q4vo). The authors also addressed scalability concerns by showing that the framework scales efficiently and remains robust even when the pool of available methods is doubled or when the most-used methods are removed (okKY).

* Thorough Rebuttal: The authors provided comprehensive responses and new experiments to address all questions. This included ablations on the sensitivity to estimation samples (okVb), analysis of computational costs (okVb), comparisons to new baselines (v8TG), and robustness checks (okKY), which satisfied all reviewers.

This is a strong paper with clear contributions to an important problem. The combination of theoretical insight and strong empirical validation makes it an excellent fit for NeurIPS. All reviewers (v8TG, okVb, okKY, Q4vo) recommend acceptance.